# PRD-2 directly regulates *casein kinase I* and counteracts nonsense-mediated decay in the Neurospora circadian clock

Christina M Kelliher[1], Randy Lambreghts[1], Qijun Xiang[1], Christopher L Baker[1,2], Jennifer J Loros[3], Jay C Dunlap[1]*

[1]Department of Molecular & Systems Biology, Geisel School of Medicine at Dartmouth, Hanover, United States; [2]The Jackson Laboratory, Bar Harbor, United States; [3]Department of Biochemistry & Cell Biology, Geisel School of Medicine at Dartmouth, Hanover, United States

**Abstract** Circadian clocks in fungi and animals are driven by a functionally conserved transcription–translation feedback loop. In *Neurospora crassa*, negative feedback is executed by a complex of Frequency (FRQ), FRQ-interacting RNA helicase (FRH), and casein kinase I (CKI), which inhibits the activity of the clock's positive arm, the White Collar Complex (WCC). Here, we show that the *prd-2* (*period-2*) gene, whose mutation is characterized by recessive inheritance of a long 26 hr period phenotype, encodes an RNA-binding protein that stabilizes the *ck-1a* transcript, resulting in CKI protein levels sufficient for normal rhythmicity. Moreover, by examining the molecular basis for the short circadian period of *upf-1*[*prd-6*] mutants, we uncovered a strong influence of the Nonsense-Mediated Decay pathway on CKI levels. The finding that circadian period defects in two classically derived Neurospora clock mutants each arise from disruption of *ck-1a* regulation is consistent with circadian period being exquisitely sensitive to levels of *casein kinase I.*

*For correspondence:
jay.c.dunlap@dartmouth.edu

Competing interests: The authors declare that no competing interests exist.

## Introduction

The Neurospora circadian oscillator is a transcription–translation feedback loop that is positively regulated by the White Collar Complex (WCC) transcription factors, which drive expression of the negative arm component Frequency (FRQ). In this way the fungal core circadian oscillator shares a common regulatory architecture with the mammalian core clock. In Neurospora, the circadian negative arm complex is composed of FRQ and FRQ-interacting RNA helicase (FRH), which together bring casein kinase I (CKI) to promote phosphorylation of WCC on key phospho-sites to inhibit its activity (*Wang et al., 2019*). FRQ is extensively regulated transcriptionally, translationally, and post-translationally over the circadian day leading ultimately to its inactivation (reviewed in: *Hurley et al., 2016*).

Indeed, in both animals and fungi, the negative arm components are regulated at the RNA and protein levels to maintain circadian phase and period, and many of the molecular details of this regulation, the focus of this paper, are conserved. Negative arm components FRQ and PER are regulated by anti-sense transcription (*Koike et al., 2012*; *Kramer et al., 2003*), by thermally regulated splicing (*Colot et al., 2005*; *Majercak et al., 1999*), and display characteristics of intrinsically disordered proteins (*Pelham et al., 2020*). Another highly conserved feature of fungal, insect, and mammalian negative arm components is progressive phosphorylation leading to their inactivation (*Baker et al., 2009*; *Ode et al., 2017*; *Vanselow et al., 2006*) (reviewed in: *Dunlap and Loros, 2018*). Taken together, FRQ, PERs, and CRYs are tightly regulated and underlying mechanisms are often conserved between clock models despite evolutionary sequence divergence of these negative arm components.

In contrast, less is known about the mechanisms regulating expression of the other essential member of the negative arm complex, CKI, orthologs of which are highly conserved in sequence and in function across eukaryotic clocks. CKI forms a stable complex as FRQ–FRH–CKIa in Neurospora (*Baker et al., 2009*; *Görl et al., 2001*), as PER-DOUBLETIME (DBT) in flies (*Kloss et al., 2001*), and as a multi-protein complex of PER-CRY-CKIδ in mouse (*Aryal et al., 2017*). Fungal CKI phosphorylates both FRQ and WCC (*He et al., 2006*). Insect DBT and mammalian CKIδ/ε are key regulators of the PER2 phospho-switch, differentially phosphorylating two regions that control PER2 turnover (*Top et al., 2018*; *Zhou et al., 2015*). Thus, CKI phosphorylations contribute to feedback loop closure in all species. FRQ–CKI binding strength is a key regulator of period length and an important oscillator variable first described in Neurospora (*Liu et al., 2019*). CKI abundance is not rhythmic in any species described to date (*Görl et al., 2001*; *Kloss et al., 2001*), but preliminary evidence suggests that its expression levels are tightly controlled to keep the clock on time, just like FRQ/PER/CRY. In mammals, CKI knockdown or knockout significantly lengthens period (*Isojima et al., 2009*; *Lee et al., 2009*; *Tsuchiya et al., 2016*), and CKIδ levels are negatively regulated by m6A methylation (*Fustin et al., 2018*). In Neurospora, decreasing the amounts of the *casein kinase I* (*ck-1a*) transcript using a regulatable promoter leads to long period defects up to ~30 hr (*Mehra et al., 2009*). CKI has a conserved C-terminal domain involved in autophosphorylation and inhibition of kinase activity (*Gietzen and Virshup, 1999*; *Guo et al., 2019*). Fungal mutants lacking this CKI C-terminal inhibitory domain have hyperactive kinase activity (*Querfurth et al., 2007*). Across clock models, the circadian period is sensitive to CKI abundance and activity due to its importance in circadian feedback loop closure.

Our modern understanding of the circadian clock was founded on genetic screens and characterization of mutants with circadian defects (*Feldman and Hoyle, 1973*; *Konopka and Benzer, 1971*; *Ralph and Menaker, 1988*). The fungal clock model *Neurospora crassa* has been a top producer of relevant circadian mutants due to its genetic tractability, ease of circadian readout, and functional conservation with the animal circadian clock (reviewed in: *Loros, 2020*). Forward genetic screens used the *ras-1$^{bd}$* mutant background (which forms distinct bands of conidiophores once per subjective night) in race tube (RT) assays to identify key players in the circadian clock (*Belden et al., 2007*; *Feldman and Hoyle, 1973*; *Sargent et al., 1966*). Genetic epistasis among the *period* genes, and in some cases, genetic mapping of mutations was also performed using *N. crassa* (*Feldman and Hoyle, 1976*; *Gardner and Feldman, 1981*; *Morgan and Feldman, 2001*). The *period* (*prd*) mutants in Neurospora are distinct from the *Drosophila* gene *period* (*per*) (*Konopka and Benzer, 1971*) and its mammalian orthologs.

All but one of the extant *period* genes in Neurospora have been cloned, and their identities have expanded our knowledge of core-clock modifying processes. *prd-4* (*period-4*), encoding checkpoint kinase 2 (Chk2), links the clock to cell-cycle progression (*Pregueiro et al., 2006*). *prd-3* (*period-3*), encoding casein kinase II (CKII), implicated direct phosphorylation of core clock proteins as central to temperature compensation (*Mehra et al., 2009*). *prd-1* (*period-1*) encodes an essential RNA-helicase that regulates the core clock under high nutrient environments (*Emerson et al., 2015*). *prd-6* (*period-6*, hereafter referred to as *upf1$^{prd-6}$*) encodes the core UPF1 subunit of the Nonsense-Mediated Decay (NMD) complex (*Compton, 2003*), although its circadian role remains cryptic. Among the available *prd* genes, only *prd-2* (*period-2*) remains uncharacterized.

We have mapped the *prd-2* mutation to NCU01019 using whole genome sequencing, and discovered its molecular identity; however, attributing its long period mutant phenotype to molecular function has remained elusive (*Lambreghts, 2012*). Equipped with the identity of PRD-2, we then followed up on the observation that the *upf1$^{prd-6}$* short period phenotype is completely epistatic to the *prd-2* mutant's long period (*Morgan and Feldman, 1997*; *Morgan and Feldman, 2001*). We find that UPF1$^{PRD-6}$ and PRD-2 use distinct mechanisms to play opposing roles in regulating levels of the *casein kinase I* transcript in Neurospora, thus rationalizing the circadian actions of the two clock mutants whose roles in the clock were not understood. PRD-2 stabilizes the *ck-1a* mRNA transcript, and the clock-relevant domains and biochemical evaluation of the PRD-2 protein indicate that it acts as an RNA-binding protein. We genetically rescue the long period phenotype of *prd-2* mutants by expressing a hyperactive CKI allele and by titrating up *ck-1a* mRNA levels using a regulatable promoter. The endogenous *ck-1a* transcript has a strikingly long 3'-UTR, indicating that its mRNA could be subject to NMD during a normal circadian day. We confirm that *upf1$^{prd-6}$* mutants have elevated levels of *ck-1a* in the absence of NMD, and further rescue the short period defect of *upf1$^{prd-6}$*

mutants by titrating down *ck-1a* mRNA levels using an inducible promoter. Taken together, a unifying model emerges to explain the action of diverse *period* mutants, where the *casein kinase I* transcript is subject to complex regulation by NMD and an RNA-binding protein, PRD-2, to control its gene expression and maintain a normal circadian period.

## Results

### An interstitial inversion identifies *prd-2*

Genetic mapping and preliminary analyses identified *prd-2* as a recessive mutant with an abnormally long ~26 hr period length that mapped to the right arm of LG V (*Morgan and Feldman, 1997*; *Morgan and Feldman, 2001*). Genetic fine structure mapping using selectable markers flanking *prd-2*, in preparation for an anticipated chromosome walk, revealed an extensive region of suppressed recombination in the region of the gene, consistent with the existence of a chromosome inversion (*Lambreghts, 2012*). PCR data consistent with this prompted whole genome sequencing that revealed a 322 kb inversion on chromosome V (*Lambreghts, 2012*) in the original isolate strain hereafter referred to as *prd-2^INV^*. The left breakpoint of the inversion occurs in the 5'-UTR of NCU03775, and its upstream regulatory sequences are displaced in the *prd-2^INV^* mutant. However, a knockout of NCU03775 (FGSC12475) has a wild-type circadian period length, unlike the long period *prd-2^INV^* mutant (*Figure 1—figure supplement 1*). The next closest gene upstream of the left inversion is NCU03771, but its transcription start site (TSS) is >7 kb away. The right breakpoint of the inversion occurs in the 5'-UTR of NCU01019, disrupting 333 bases of its 5'-UTR and its entire promoter region (*Figure 1A and B*). A knockout of NCU01019 has a 26 hr long period, matching the *prd-2^INV^* long period phenotype (*Figure 1C*). The *prd-2^INV^* mutant has drastically reduced levels of NCU01019 gene expression in constant light conditions and in the subjective evening of a circadian free run (*Figure 1D*), suggesting that the inversion completely disrupts the NCU01019 promoter and TSS. Placing NCU01019 under the nutrient-responsive *qa-2* promoter, we find that the long period length occurs at very low gene expression levels using $10^{-6}$ M quinic acid induction (*Figure 1E*). Finally, ectopic expression of NCU01019 at the *csr-1* locus in the *prd-2^INV^* background rescues the long period phenotype (*Figure 1F*). We conclude that PRD-2 is encoded by NCU01019.

We mapped the clock-relevant domains of the PRD-2 protein (*Figure 2A*), finding that both an SUZ domain and the proline-rich C-terminus of PRD-2 are required for a normal clock period. This result was confirmed in two separate genetic backgrounds either by replacing the endogenous locus with domain deletion mutants (*Figure 2B*) or by ectopic expression of domain mutants at the *csr-1* locus in a Δ*prd-2* background (*Figure 2C*; *Supplementary file 1*). The SUZ domain family can bind RNA directly in vitro (*Song et al., 2008*), but curiously PRD-2's adjacent R3H domain, which is better characterized in the literature as a conserved RNA-binding domain, is dispensable for clock function. The C-terminus of PRD-2 is predicted to be highly disordered, and finer mapping of this region showed that neither a glutamine/proline-rich domain (amino acids 525–612, 21% Gln, 26% Pro) nor a domain conserved across fungal orthologs (amino acids 625–682, 21% Pro) were required for normal clock function (*Figure 2C*). The remainder of the C-terminus (amino acids 495–524, 28% Pro; 683–790, 24% Pro) contains a clock-relevant region of PRD-2 based on deletion analyses. Further, PRD-2 SUZ domain and C-terminal deletion mutants are expressed at the protein level, indicating that clock defects must be due to the absent domain (*Figure 2—figure supplement 1A*). PRD-2 is exclusively localized to the cytoplasm based on biochemical evaluation, and this localization does not change as a function of time of day (*Figure 2D*).

NCU01019 RNA expression is not induced by light (*Wu et al., 2014*) nor rhythmically expressed over circadian time (*Hurley et al., 2014*). NCU01019 protein is abundant and shows weak rhythms (*Hurley et al., 2018*; *Figure 2—figure supplement 1B*), which suggests that PRD-2 oscillations are driven post-transcriptionally to peak in the early subjective morning, prior to the peak in the *frq* transcript (*Aronson et al., 1994*). Rhythms in PRD-2 protein expression were confirmed using a luciferase translational fusion (*Figure 2—figure supplement 1C*), which peaked during the circadian day. *prd-2^INV^* and ΔNCU01019 have a slight growth defect (*Figure 1C*) and are less fertile than wild type as the female partner in a sexual cross (data not shown). Temperature and nutritional compensation of ΔNCU01019 alone are normal (*Figure 2—figure supplement 2*), which was expected given the normal TC profile of the *prd-2^INV^* mutant (*Gardner and Feldman, 1981*). PRD-2 (XP_961631.1) is

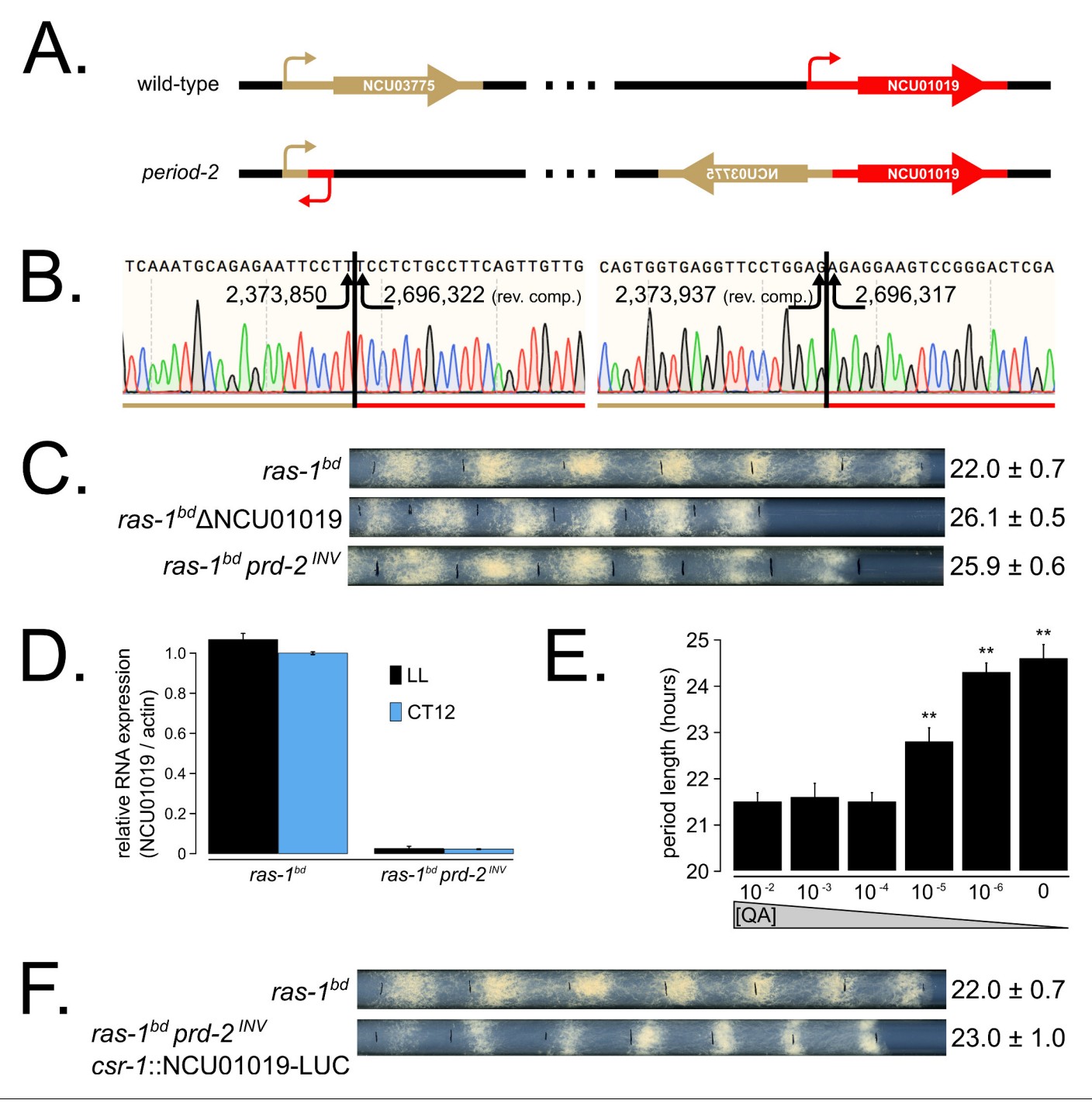

**Figure 1.** The *prd-2* phenotype derives from reduced expression of NCU01019. Whole genome sequencing identified a 322,386 bp inversion on linkage group V in the original *prd-2* mutant strain (**Lambreghts, 2012**). The inversion breakpoints disrupt two loci, NCU03775 and NCU01019, depicted in cartoon form (**A**). Sanger sequencing confirms the DNA sequence of the left and right breakpoints, and the corresponding NC12 genome coordinates are shown at each arrowhead (**B**). Circadian period length was determined by race tube (RT) assay for *ras-1^{bd}* controls, targeted deletion of the NCU01019 locus, and the classically derived *prd-2^{INV}* mutant. The ΔNCU01019 mutant has a long period and slow growth defect similar to *prd-2^{INV}* (**C**). NCU01019 RNA expression levels are detectable by RT-qPCR in the *prd-2^{INV}* mutant but are drastically reduced compared to *ras-1^{bd}* controls grown in constant light (LL) or at subjective dusk (CT12) during a circadian free run (**D**). After replacing the endogenous promoter of NCU01019 with the inducible *qa-2* promoter, addition of high levels of quinic acid ($10^{-2}$ to $10^{-3}$ M) led to a normal circadian period by RT assay ($10^{-2}$ M τ = 21.5 ± 0.2 hr; $10^{-3}$ M τ = 21.6 ± 0.3 hr; $10^{-4}$ M τ = 21.5 ± 0.2 hr). Lower levels of QA inducer led to a long circadian period ($10^{-5}$ M τ = 22.8 ± 0.3 hr; $10^{-6}$ M τ = 24.3 ± 0.2 hr; 0 QA τ = 24.6 ± 0.3 hr) due to reduced NCU01019 expression. Asterisks (**) indicate p<1 × $10^{-10}$ by Student's t-test compared to

*Figure 1 continued on next page*

*Figure 1 continued*

$10^{-2}$ M QA RT results (E). The entire NCU01019 locus (plus 951 bases of its upstream promoter sequence) was fused in-frame with codon-optimized luciferase. Ectopic expression of this NCU01019-luc construct in the *prd-2^INV* background rescues the long period phenotype by RT assay (F).

The online version of this article includes the following figure supplement(s) for figure 1:

**Figure supplement 1.** NCU03775 knockout has a normal circadian period and does not explain the *prd-2^INV* phenotype.

well conserved among Ascomycota fungi as noted by BLASTp scores (<e-70), while only its R3H and/or SUZ domains have significant similarity to insect and mammalian proteins: the *encore* gene in flies and the *R3HDM1*, *R3HDM2*, and *ARPP21* genes in human and mouse.

## PRD-2 regulates CKI levels

To identify the putative mRNA targets of PRD-2, we performed total RNA-sequencing on triplicate samples of Δ*prd-2* versus control grown in constant light at 25°C. Hundreds of genes are affected by loss of PRD-2, but we did not identify a consensus functional category or sequence motif(s) for the putative PRD-2 regulon (*Figure 3—figure supplement 1*). Given the pleotropic phenotypes of Δ*prd-2*, we posit that PRD-2 plays multiple roles in the cell, including regulation of carbohydrate and secondary metabolism. Focusing specifically on core clock genes, we found that *ck-1a*, *frq*, *wc-2*, *ckb-1* (regulatory beta subunit of CKII), and *frh* were significantly altered in the absence of PRD-2 (*Figure 3A*). Pursuing the top two hits, we found that the CKI transcript was dramatically less stable in Δ*prd-2* (*Figure 3B*), while *frq* mRNA stability was not significantly altered (*Figure 3—figure supplement 2*). To demonstrate that PRD-2 binds the *ck-1a* transcript in vivo, we used RNA immunoprecipitation after UV cross-linking (CLIP). The Pumilio family RNA-binding protein PUF4 (NCU16560) was previously shown to bind in the 3'-UTR of *cbp3* (NCU00057), *mrp-1* (NCU07386), and other target genes identified by HITS-CLIP high-throughput sequencing (*Wilinski et al., 2017*). C-terminally tagged alleles of PRD-2, PUF4, and an untagged negative control strain were used to immunoprecipitate cross-linked RNAs (Materials and methods). As expected, *cbp3* and *mrp-1* positive controls were significantly enriched in the PUF4 CLIP sample compared to the negative IP (*Figure 3C*). *ck-1a* is also enriched in the PRD-2 CLIP sample, demonstrating that the CKI transcript is a direct target of the PRD-2 protein (*Figure 3C*).

Hypothesizing that the clock-relevant target of PRD-2 could be CKI, we used two genetic approaches to manipulate CKI activity in an attempt to rescue the Δ*prd-2* long period phenotype. First, we placed the *ck-1a* gene under the control of the quinic acid inducible promoter (*Mehra et al., 2009*) and crossed this construct into the Δ*prd-2* background. We found that increasing expression of *ck-1a* using high levels ($10^{-1}$ to $10^{-2}$ M) of QA partially rescued the Δ*prd-2* long period phenotype (*Figure 4A*). We also noticed a synergistic poor growth defect in the double mutant at $10^{-4}$ M QA, consistent with low levels of *ck-1a* (an essential gene in Neurospora: *Görl et al., 2001*; *He et al., 2006*). There are two explanations for the lack of full rescue to periods shorter than 25 hr in the $P_{qa-2}$-*ck-1a* Δ*prd-2* double mutant: (1) even at saturating $10^{-1}$ M QA induction, the *qa-2* promoter may not reach endogenous levels of *ck-1a* achieved under its native promoter, and/or (2) because PRD-2 acts directly as an RNA-binding protein for CKI transcripts, simply increasing levels of *ck-1a* RNA cannot fully rescue PRD-2's role in stabilizing or positioning CKI transcripts in the cytoplasm.

Next, we turned to a previously described fungal CKI constitutively active allele, CKI Q299^STOP (*Querfurth et al., 2007*), reasoning that we might be able to rescue low *ck-1a* levels in Δ*prd-2* by genetically increasing CKI kinase activity. We replaced endogenous CKI with a CKI^SHORT allele, which expresses only the shortest *ck-1a* isoform (361 amino acids). CKI^SHORT lacks 23 amino acids in the C-terminal tail of the full length isoform that are normally subject to autophosphorylation leading to kinase inhibition. This CKI^SHORT allele also carries an in-frame C-terminal HA3 tag and selectable marker, which displace the endogenous 3'-UTR of *ck-1a*. The CKI^SHORT mutant has a short period phenotype (~17 hr), presumably due to hyperactive kinase activity and rapid feedback loop closure (*Liu et al., 2019*). Significantly, the CKI^SHORT mutation is completely epistatic to Δ*prd-2* (*Figure 4B*), indicating that CKI is the clock-relevant target of PRD-2.

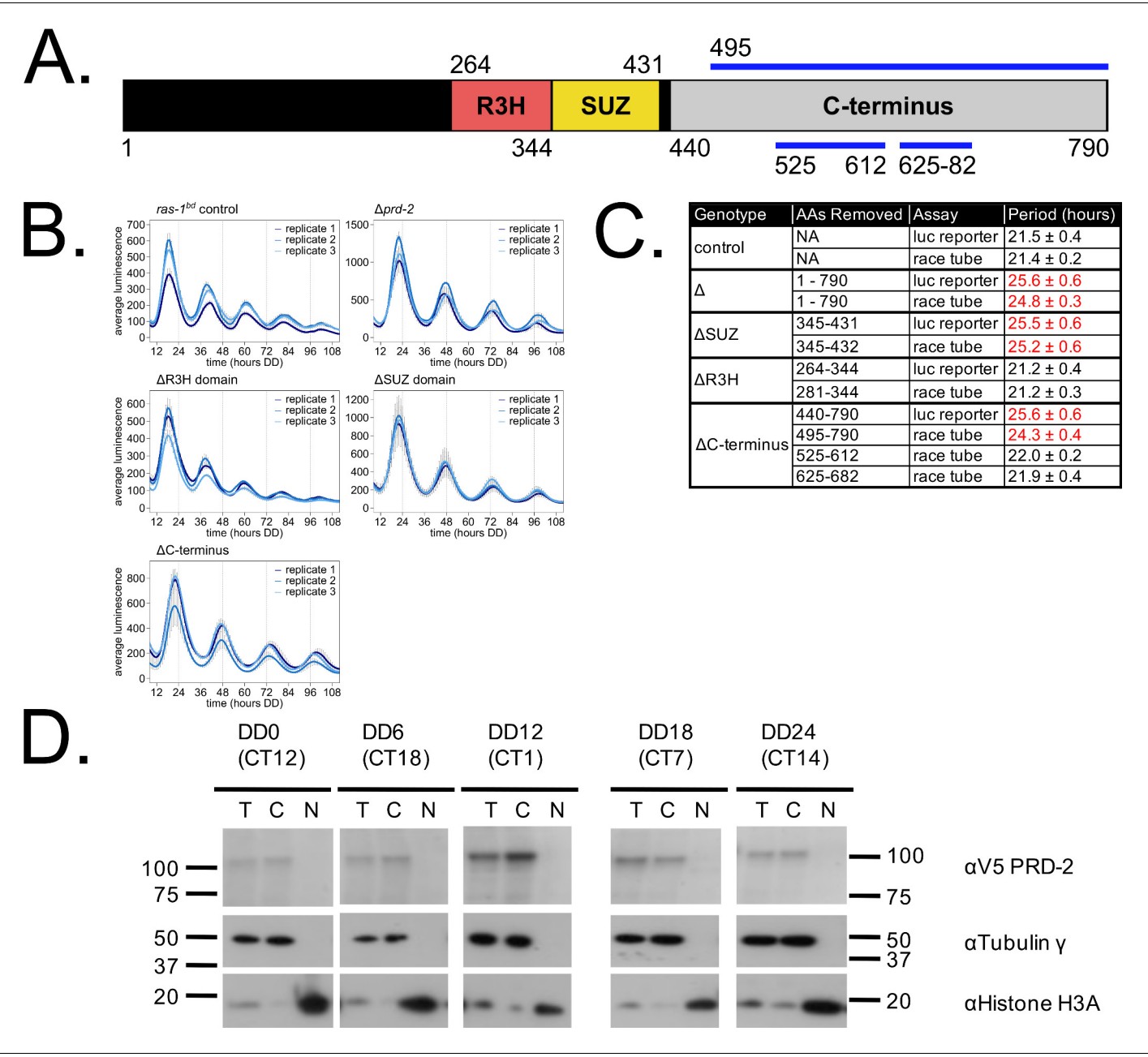

**Figure 2.** Clock-relevant protein domains and localization of PRD-2 suggest and RNA-binding function. PRD-2 has tandemly arrayed R3H and SUZ domains associated with RNA binding proteins, and its C-terminal region is highly enriched for proline (P) and glutamine (Q). The cartoon of PRD-2 protein lists relevant amino acid coordinates (**A**). The native NCU01019 locus was replaced with single domain deletion mutants, and 96-well plate luciferase assays were used to measure the circadian period length in triplicate wells per biological replicate experiment. A wild-type clock period was recovered in *ras-1^{bd}* controls and the *prd-2*ΔR3H mutant, while Δ*prd-2*, *prd-2*ΔSUZ, and *prd-2*ΔC-terminus had long period phenotypes (**B**). Independently constructed strains targeted domain deletion mutants to the *csr-1* locus in a Δ*prd-2* background (**Supplementary file 1**), and mutant period lengths were determined by race tube assay. Period lengths (±1 SD) show that the clock-relevant domains of PRD-2 are the SUZ domain and the C-terminus (**C**). Total (T), Nuclear (N), and Cytosolic (C) fractions were prepared over a circadian time course (N = 1 per time point). γ-Tubulin (NCU03954) was used as a control for cytoplasmic localization and histone H3 (NCU01635) for nuclear localization. PRD-2 tagged with a C-terminal V5 epitope tag is localized to the cytoplasm throughout the circadian cycle (**D**).

The online version of this article includes the following figure supplement(s) for figure 2:

**Figure supplement 1.** PRD-2 protein levels are slightly rhythmic and are detectable in protein domain deletion mutants.
**Figure supplement 2.** Temperature and nutritional compensation are normal in ΔNCU01019.

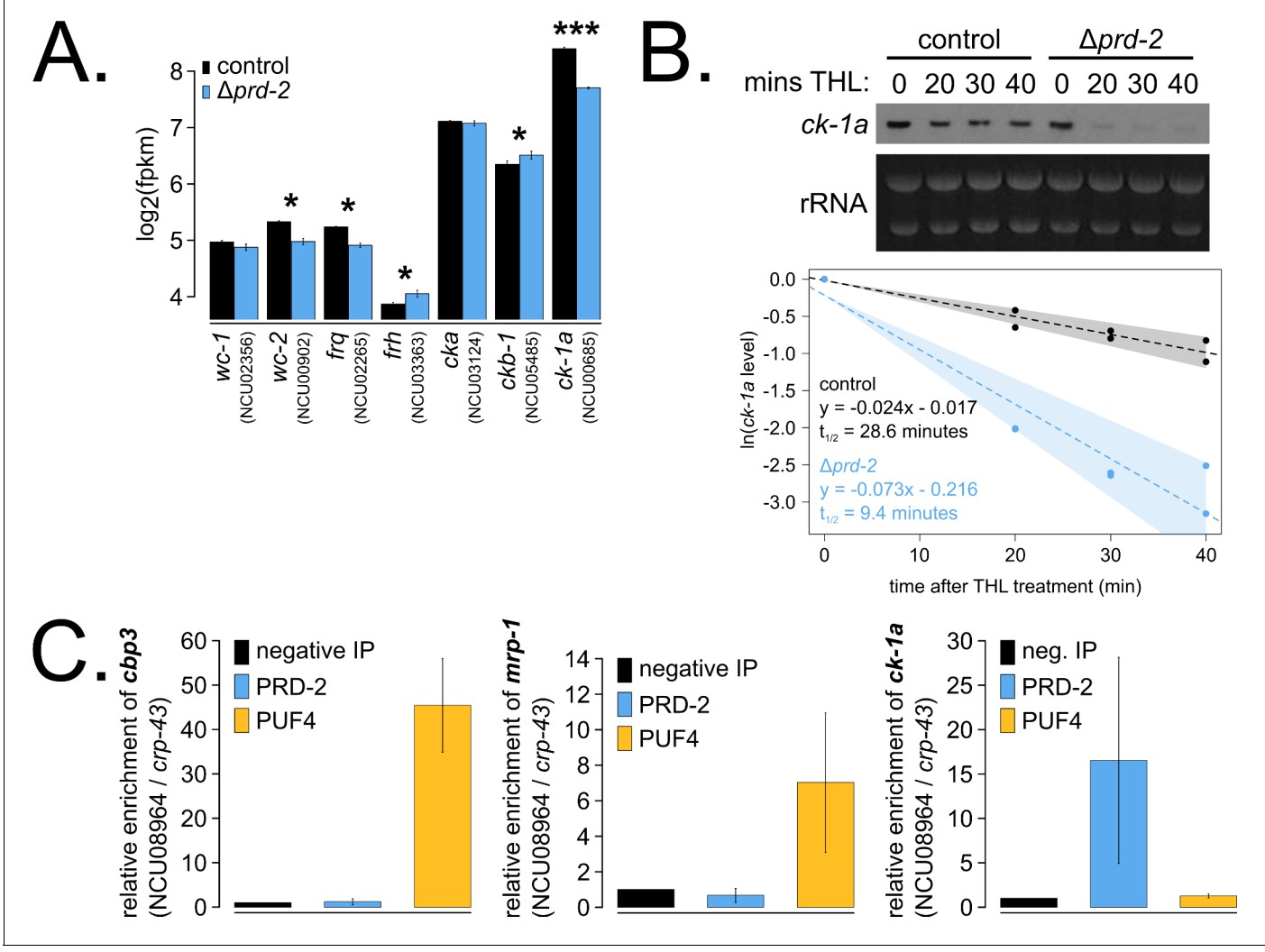

**Figure 3.** The core clock target of PRD-2 is the *casein kinase I* transcript. Control and Δ*prd-2* cultures were grown in the light at 25°C in Bird medium for 48 hr prior to RNA isolation. Expression levels for core clock genes were measured by RNA-sequencing (N = 3 biological replicates per strain), and log₂-transformed FPKM values are shown. Asterisks indicate $p<0.05$ (*) or $p<5 \times 10^{-5}$ (***) by Student's t-test compared to control levels. The *ck-1a* transcript is >1.5× less abundant in Δ*prd-2* (A). *ck-1a* mRNA degradation kinetics were examined by Northern blot in a time course after treatment with thiolutin (THL) at approximately CT1 (N = 2 biological replicates). RNA levels were quantified using ImageJ, natural log transformed, fit with a linear model (glm in R, Gaussian family defaults), and half-life was calculated assuming first order decay kinetics (ln(2)/slope). Shaded areas around the linear fit represent 95% confidence intervals on the slope. The *ck-1a* transcript is 3× less stable in Δ*prd-2* (B). The PUF4 (NCU16560) RNA-binding protein pulls down known target transcripts *cbp3* (NCU00057) and *mrp-1* (NCU07386) by RT-qPCR (N = 3 biological replicates). PRD-2 CLIP samples were processed in parallel with PUF4 positive controls, and PRD-2 binds the *ck-1a* transcript in vivo (C).

The online version of this article includes the following figure supplement(s) for figure 3:

**Figure supplement 1.** Hundreds of genes have altered expression levels in the Δ*prd-2* mutant but a common pathway or sequence motif was not detected.

**Figure supplement 2.** Loss of *prd-2* has little effect on stability of the *frq* transcript.

## NMD impacts the clock by regulating CKI levels

NMD in *Neurospora crassa* is triggered by various mRNA signatures. Open reading frames in 5'-UTRs that produce short peptides (5'-uORFs) can trigger NMD in a mechanism that does not require the Exon Junction Complex (EJC; *Zhang and Sachs, 2015*). The *frq* transcript has six such uORFs (*Colot et al., 2005*; *Diernfellner et al., 2005*) and could be a bona fide NMD target because its splicing is disrupted in the absence of NMD (*Wu et al., 2017*). Transcripts containing long 3'-UTRs are also subject to NMD regulation. In addition, transcripts with intron(s) near a STOP codon and/or

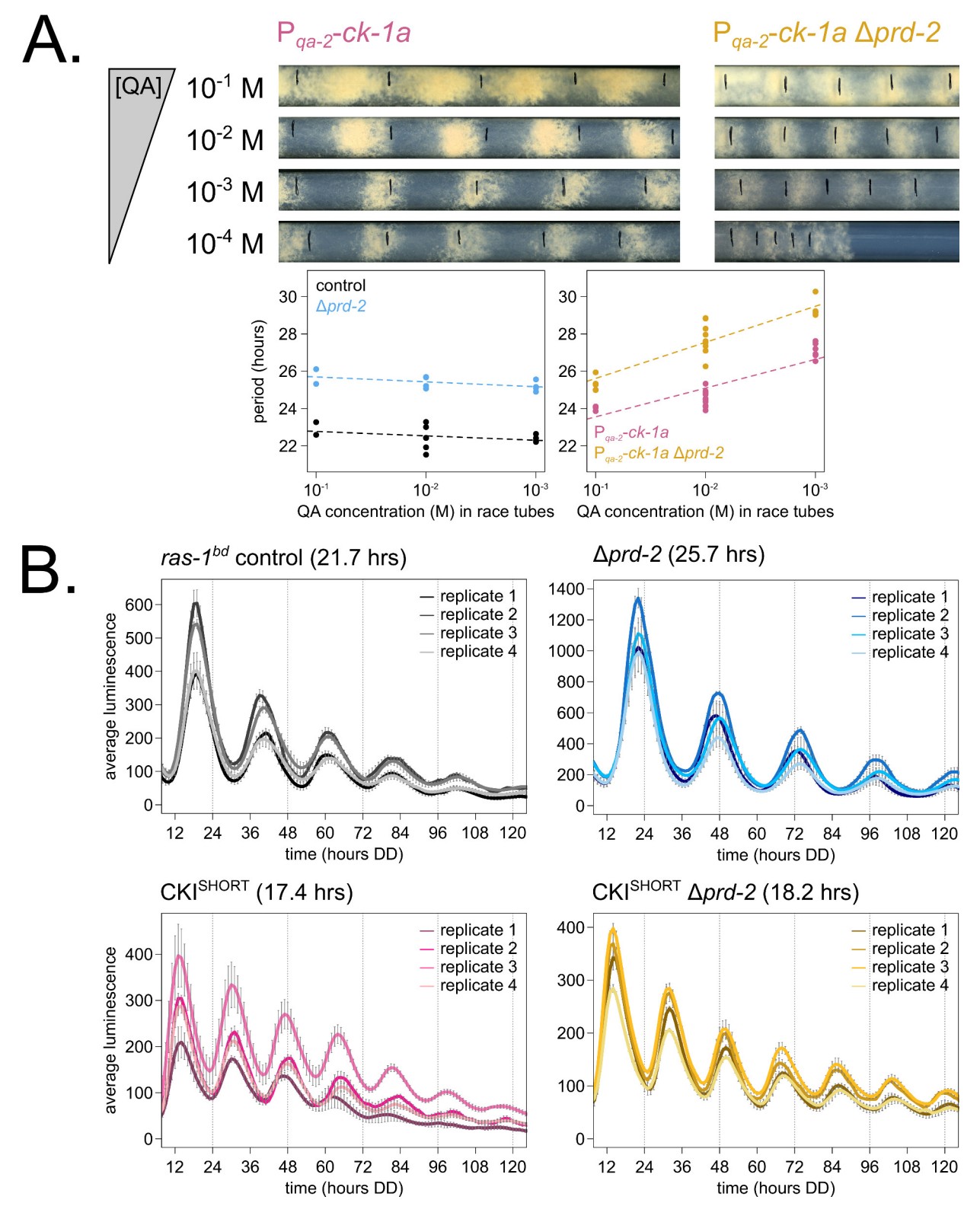

**Figure 4.** Genetically increasing casein kinase I (CKI) levels or activity rescues the Δ*prd-2* long period phenotype. Representative race tubes (RTs) from *ras-1bd* P*qa-2*-*ck-1a* single (pink) and *ras-1bd* P*qa-2*-*ck-1a* Δ*prd-2* double (yellow) mutants are shown with growth using the indicated concentrations of quinic acid (QA) to drive expression of *ck-1a*. All results are shown in a scatterplot, where each dot represents one RT's free running period length. *ras-1bd* controls (black) had an average period of 22.5 ± 0.5 hr (N = 12), and period length was not significantly affected by QA concentration (ANOVA

*Figure 4 continued on next page*

*Figure 4 continued*

p=0.297). *ras-1^bd Δprd-2* controls (blue) had an average period of 25.4 ± 0.4 hr (N = 10), and period length was not significantly affected by QA concentration (ANOVA p=0.093). Period length of *ras-1^bd* P*~qa-2~-ck-1a* single mutants (pink) was significantly altered across QA levels (ANOVA p=3.6 × 10^{-6}), and the average period at 10^{-1} M QA was 24.3 ± 0.5 hr (N = 4). Period length of *ras-1^bd* P*~qa-2~-ck-1a Δprd-2* double mutants (yellow) was also significantly affected by QA levels (ANOVA p=8.1 × 10^{-8}), and the average period at 10^{-1} M QA was 25.4 ± 0.4 hr (N = 4). The double mutant period length was not genetically additive at high levels of QA induction (**A**). A hyperactive CKI allele was constructed by expressing the shortest isoform only (CKI^{SHORT}). 96-well plate luciferase assays were used to measure the circadian period length. Traces represent the average of three technical replicates across four biological replicate experiments for: *ras-1^bd* controls (gray, τ = 21.7 ± 0.3 hr), *ras-1^bd Δprd-2* (blue, τ = 25.7 ± 0.6 hr), *ras-1^bd* CKI^{SHORT} (pink, τ = 17.4 ± 0.3 hr), and *ras-1^bd* CKI^{SHORT} *Δprd-2* double mutants (yellow, τ = 18.2 ± 0.3). CKI^{SHORT} is completely epistatic to *Δprd-2* in double mutants (**B**).

with intron(s) in the 3'-UTR can be degraded by NMD after recruitment of the UPF1/2/3 complex by the EJC in a pioneering round of translation (*Zhang and Sachs, 2015*).

Since the observation by *Compton, 2003* that the short period mutant *prd-6* identified the UPF1 core subunit of the NMD pathway, the clock-relevant target(s) of NMD has been an object of conjecture and active research. Because loss of NMD reduces the amount of the transcript encoding the short-FRQ protein isoform (*Wu et al., 2017*), and strains making only short-FRQ have slightly lengthened periods (*Liu et al., 1997*), *Wu et al., 2017* recently speculated that the short period of the *upf1^{prd-6}* mutant might be explained by effects of NMD on FRQ. However, strains expressing only long-FRQ display an essentially wild-type period length (*Colot et al., 2005*; *Liu et al., 1997*), not a short period phenotype like *upf1^{prd-6}*; this finding is not consistent with FRQ being the only or even principal clock-relevant target of NMD, leaving unresolved the role of NMD in the clock.

To tackle this puzzle, we returned to classical genetic epistasis experiments and confirmed the observation that *upf1^{prd-6}* is completely epistatic to *prd-2^{INV}* (*Morgan and Feldman, 2001*), going on to show that in fact each of the individual NMD subunit knockouts, *Δupf2* and *Δupf3* as well as *Δupf1^{prd-6}*, is epistatic to the *Δprd-2* long period phenotype (*Figure 5A*). Previous work had profiled the transcriptome of *Δupf1^{prd-6}* compared to a control (*Wu et al., 2017*); we re-processed this RNA-seq data and found, exactly as in *Δprd-2*, that *ck-1a* was the most affected core clock gene in *Δupf1^{prd-6}* (*Figure 5B*). The *ck-1a* transcript has an intron located 70 nt away from its longest isoform's STOP codon, and its 3'-UTR is, remarkably, among the 100 longest annotated UTRs in the entire Neurospora transcriptome (*Figure 5C*). NMD targeting to long 3'-UTR transcripts like *ck-1a* is thought to occur independently of the EJC and nuclear cap-binding complex (CBC) in *Neurospora crassa* (*Zhang and Sachs, 2015*). We used the knockout mutant *Δcbp80* (NCU04187) to confirm that Neurospora CBC is not required for a normal circadian clock and that the long period length of *Δprd-2* is unchanged in the *Δcbp80* background (*Figure 5—figure supplement 1*). Thus, *ck-1a* is a strong candidate for NMD-mediated degradation via its long 3'-UTR, not dependent on EJC and CBC components.

We hypothesized that CKI is overexpressed in the absence of NMD (*Figure 5B*), leading to faster feedback loop closure and a short circadian period. To genetically control *ck-1a* levels, we crossed the regulatable P*~qa-2~-ck-1a* allele into the *Δupf1^{prd-6}* background and confirmed our hypothesis by finding that at low levels of inducer (10^{-5} M QA), decreased levels of *ck-1a* transcript revert the short period length of *Δupf1^{prd-6}* to control period lengths (*Figure 5D*). Further, protein levels of CKI in the *Δupf1^{prd-6}* background are reduced to control levels at 10^{-5} M QA (*Figure 5E*), which explains the period rescue phenotype. CKI protein is two to three times more abundant in *Δupf1^{prd-6}* and in *Δprd-2 Δupf1^{prd-6}* (*Figure 5F*), matching its overexpression in the *Δupf1^{prd-6}* transcriptome (*Figure 5B*). CKI protein is 3× reduced in *Δprd-2* (*Figure 5F*), also correlating with its reduced mRNA expression and stability (*Figure 3*). We conclude that CKI is also the clock-relevant target of UPF1^{PRD-6}, placing NMD, PRD-2, and CKI in the same genetic epistasis pathway.

## Discussion

By uncovering the identity and mode of action of PRD-2 and exploring the mechanism of two classical *period* mutants, *prd-2* and *upf1^{prd-6}*, we found a common basis in regulation of CKI levels, which are under tight control in the Neurospora clock (*Figure 6*). That the mechanistic basis of action of two independently derived non-targeted clock mutants centers on regulation of the activity of a

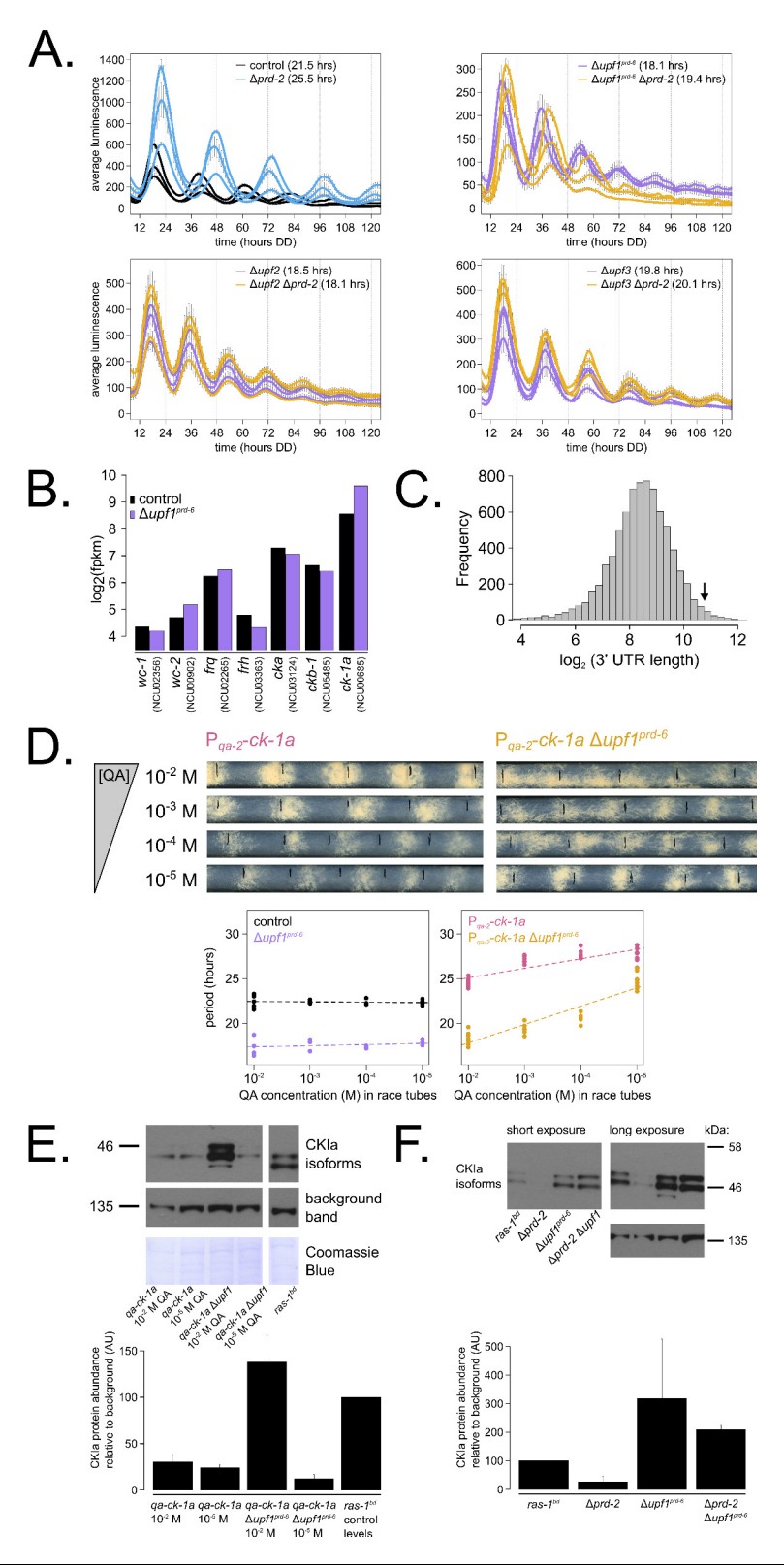

**Figure 5.** Nonsense-mediated decay (NMD) negatively regulates casein kinase I (CKI) levels via UPF1[PRD-6], establishing a basis for the *upf1[prd-6] prd-2* genetic epistasis on circadian period length. 96-well plate luciferase assays were used to measure the circadian period length in triplicate wells per three biological replicate experiments for: *ras-1[bd]* controls (black, τ = 21.5 ± 0.3 hr), *ras-1[bd]* Δ*prd-2* (blue, τ = 25.5 ± 0.4 hr); *ras-1[bd]* Δ*upf1[prd-6]* (purple, τ = 18.1 ± 0.2 hr), *ras-1[bd]* Δ*upf1[prd-6]*Δ*prd-2* double mutants (yellow, τ = 19.4 ± 0.7 hr); *ras-1[bd]* Δ*upf2* (purple, τ = 18.5 ± 0.5 hr), *ras-1[bd]* Δ*upf2* Δ*prd-2*

*Figure 5 continued on next page*

*Figure 5 continued*

double mutants (yellow, τ = 18.1 ± 0.3 hr); *ras-1*^bd^ Δ*upf3* (purple, τ = 19.8 ± 0.3 hr), *ras-1*^bd^ Δ*upf3* Δ*prd-2* double mutants (yellow, τ = 20.1 ± 0.2 hr). Each individual NMD subunit knockout is epistatic to the Δ*prd-2* long period phenotype (**A**). Raw RNA-seq data from a previous study (***Wu et al., 2017***) were analyzed using the same pipeline as data from ***Figure 3A*** (see Materials and methods). Control and Δ*upf1*^prd-6^ gene expression levels (log$_2$-transformed) are shown for core clock genes. The *ck-1a* transcript is >2× more abundant in Δ*upf1*^prd-6^ (**B**). 3'-UTR lengths from 7793 genes were mined from the *N. crassa* OR74A genome annotation (FungiDB version 45, accessed on 10/25/2019), and plotted as a histogram. The arrow marks the 3'-UTR of *ck-1a*, which is 1739 bp and within the top 100 longest annotated UTRs in the entire genome (**C**). Representative race tubes (RTs) from *ras-1*^bd^ P$_{qa-2}$-*ck-1a* single (pink) and *ras-1*^bd^ P$_{qa-2}$-*ck-1a* Δ*upf1*^prd-6^ double (yellow) mutants are shown at the indicated concentrations of quinic acid to drive expression of *ck-1a*. All results are shown in a scatterplot, where each dot represents one RT's free running period length. *ras-1*^bd^ controls (black) had an average period of 22.4 ± 0.4 hr (N = 20), and period length was not significantly affected by QA concentration (ANOVA p=0.605). *ras-1*^bd^ Δ*upf1*^prd-6^ controls (purple) had an average period of 17.5 ± 0.6 hr (N = 16), and period length was not significantly affected by QA concentration (ANOVA p=0.362). Period length of *ras-1*^bd^ P$_{qa-2}$-*ck-1a* single mutants (pink) was significantly altered across QA levels (ANOVA p=2.9×10$^{-8}$), and the average period at 10$^{-5}$ M QA was 27.6 ± 0.8 hr (N = 8). Period length of *ras-1*^bd^ P$_{qa-2}$-*ck-1a* Δ*upf1*^prd-6^ double mutants (yellow) was also significantly affected by QA levels (ANOVA p=9.4×10$^{-12}$), and the average period at 10$^{-5}$ M QA was 24.7 ± 0.9 hr (N = 8). Thus, the double mutant period length was not genetically additive at low levels of QA induction, and the short period phenotype of Δ*upf1*^prd-6^ is rescued (**D**). CKI protein levels were measured from the indicated genotypes grown in 0.1% glucose liquid culture medium (LCM) with QA supplemented at the indicated concentrations for 48 hr in constant light. A representative immunoblot of three biological replicates is shown, and replicates are quantified in the bar graph relative to *ras-1*^bd^ control CKI levels from a 2% glucose LCM culture (**E**). CKI protein levels were measured from the indicated genotypes grown in 2% glucose LCM for 48 hr in constant light. A representative immunoblot of three biological replicates is shown, and replicates are quantified in the bar graph relative to *ras-1*^bd^ control CKI levels (**F**). CKI protein levels are increased in Δ*upf1*^prd-6^, decreased in the Δ*prd-2* mutant, and Δ*upf1*^prd-6^ is epistatic to Δ*prd-2* with respect to CKI levels and circadian period length.

The online version of this article includes the following figure supplement(s) for figure 5:

**Figure supplement 1.** The cap-binding protein CBP80 (NCU04187) is not required for a normal clock, and does not alter the Δ*prd-2* long period phenotype, suggesting that *ck-1a* degradation is controlled by NMD machinery without the Exon Junction Complex and nuclear cap-binding complex.

**Figure supplement 2.** Long untranslated regions (UTRs) are characteristic of *casein kinase I* gene orthologs across species.

single enzyme, CKI, via two distinct mechanisms is noteworthy. *prd-2* encodes an RNA-binding protein (***Figures 1*** and ***2***) that stabilizes the CKI transcript (***Figure 3B***). We demonstrate that CKI is the most important core clock target of PRD-2 by rescuing its long period mutant phenotype with a

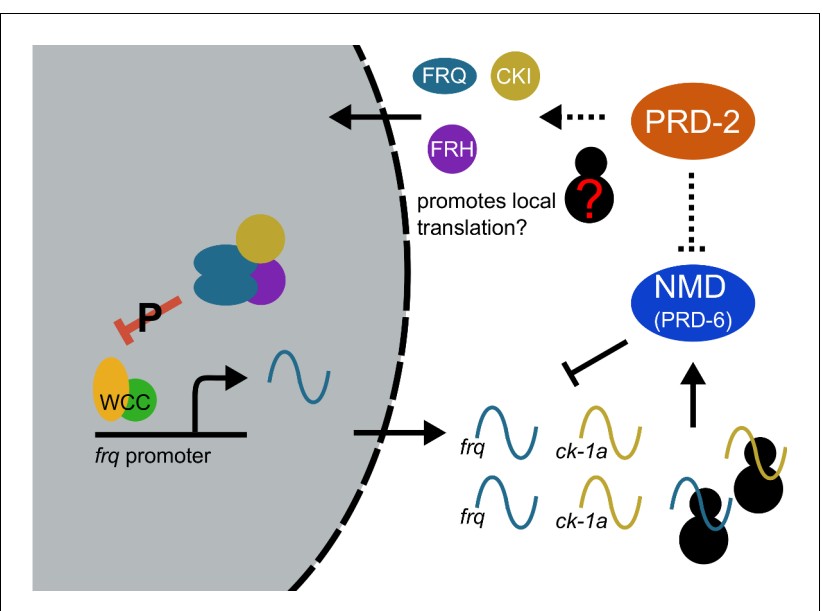

**Figure 6.** Counterbalancing regulation of casein kinase I (CKI) provides a unifying genetic model for the action of PRD-2 and UPF1^PRD-6^ in the circadian oscillator. The NMD complex (UPF1^PRD-6^, UPF2, and UPF3) targets the *frq* and *ck-1a* transcripts for degradation (upstream uORFs in *frq*; long 3'-UTR in *ck-1a*). PRD-2 binds to and stabilizes *ck-1a* transcripts (dashed lines), which could also promote local translation and complex formation for the negative arm of the clock. In the absence of PRD-2, the long period phenotype is due to low CKI levels, and in the absence of NMD, the short period phenotype is due to high CKI levels.

hyperactive CKI allele (*Figure 4B*). The predominantly cytoplasmic localization of PRD-2 (*Figure 2D*) is consistent with its action in protecting *ck-1a* transcripts from NMD and rounds out the model. PTBP1, an RNA-binding protein, protects its target transcripts from NMD-mediated degradation by binding in the 3'-UTR and blocking NMD recruitment in mouse (*Ge et al., 2016*), and future work will determine if PRD-2 functions similar to PTBP1.

This work contributes another possible example to the growing literature describing conserved post-transcriptional regulation on core clock messages. Anti-sense transcription at the *frq* locus produces the *qrf* transcript, which is required for proper phase control and light responses of the fungal clock (*Kramer et al., 2003*). The mammalian PER2 anti-sense transcript displays nearly identical dynamics to *qrf* expression (*Koike et al., 2012*). Mammalian PER2 sense expression levels are further regulated by microRNA binding sites in its 3'-UTR (*Yoo et al., 2017*). In a similar manner, *frq* RNA is directly targeted for turnover by rhythmic exosome activity in the late day (*Guo et al., 2009*). Splicing of the *frq* transcript is regulated by temperature (*Colot et al., 2005*), mirroring thermal regulation mechanisms in the clocks of *Drosophila* (*Majercak et al., 1999*) and Arabidopsis (*James et al., 2012*). The codons composing the *frq* transcript are non-optimal, which improves FRQ's co-translational folding (*Zhou et al., 2013*), and FRQ's disordered protein structure is also stabilized by its binding partner FRH (*Hurley et al., 2013*). Mammalian PER2 is also largely intrinsically disordered, and indeed circadian clock proteins across species have large stretches of intrinsic disorder which are in the early stages of functional characterization (*Pelham et al., 2020*; *Pelham et al., 2018*) (reviewed in: *Partch, 2020*). These data document the complexity of post-transcriptional regulation of clock components, and this study demonstrates that even non-rhythmic clock transcripts such as CKI are under tight regulation that is essential for normal clock function.

UPF1[PRD-6] and the NMD machinery target *ck-1a* mRNA for degradation to regulate its expression levels, presumably mediated by the long 3'-UTR of *ck-1a* transcripts in Neurospora (*Figure 5*). NMD components are not rhythmic in abundance in the fungal clock (*Hurley et al., 2014*; *Hurley et al., 2018*). These data, taken together with the constitutive expression of the CKI mRNA and protein (*Baker et al., 2009*; *Görl et al., 2001*; *Hurley et al., 2014*; *Hurley et al., 2018*), lead us to predict that NMD regulation of CKI occurs throughout the circadian cycle. To our knowledge the discovery of NMD regulation of CKI represents a wholly novel and potentially important mode of regulation for this pivotal kinase. Future work will investigate whether insect DBT and/or mammalian CKIδ/ε (CSNK1D, CSNK1E) are also targets of NMD. Long UTR length appears to be conserved across CKI orthologs (*Figure 5—figure supplement 2*). One previous study in *Drosophila* reported a circadian period defect in a tissue-specific NMD knockdown (*Ri et al., 2019*), but the behavioral rhythm was lengthened in UPF1-depleted insects unlike the short period defect observed in Neurospora. In mouse, both CKIε and CLOCK display altered splicing patterns in the absence of UPF2 (*Weischenfeldt et al., 2012*). Most core clock proteins have at least one uORF in mammals (*Millius and Ueda, 2017*), altogether raising the possibility that multiple core clock genes are regulated by NMD. The importance of NMD has already been recognized and investigated in the plant clock, where alternative splicing leads to NMD turnover for four core clock and accessory mRNAs: GRP7, GRP8, TOC1, and ELF3 (reviewed in: *Mateos et al., 2018*).

CKI abundance and alternative isoforms strongly affect circadian period length. Low levels of CKI driven from an inducible promoter lead to long periods approaching 30 hr (*Mehra et al., 2009*; *Figure 4A*). In the mammalian clock, decreased CKI expression also significantly lengthens period (*Isojima et al., 2009*; *Lee et al., 2009*; *Tsuchiya et al., 2016*). CKI is rendered hyperactive by removing its conserved C-terminal domain, a domain normally subject to autophosphorylation leading to kinase inhibition (*Gietzen and Virshup, 1999*; *Guo et al., 2019*; *Querfurth et al., 2007*). We generated a CKI mutant expressing only this shortest CKI isoform, finding a 17.5 hr short period phenotype in the absence of C-terminal autophosphorylation (*Figure 4B*). Based on prior work, increased CKI activity and/or abundance would be expected to increase FRQ-CKI affinity and lead to faster feedback loop closure (*Liu et al., 2019*), consistent with the short period phenotype. Curiously, this CKI short isoform is expressed at levels similar to the full length isoform in Neurospora (as well as a third short isoform derived from an alternative splice acceptor event) (*Figure 5F*), and all isoforms interact with FRQ by immunoprecipitation (*Querfurth et al., 2007*). Why do natural isoforms arise without the auto-inhibitory C-terminus in Neurospora, and are these regulatory events required to keep the clock on time? Mammalian alternative isoforms *CKIδ1* and *CKIδ2* have different substrate preferences in vitro, which leads to differential phosphorylation of PER2 whereby CKIδ2

phosphorylation significantly stabilizes PER2 (*Fustin et al., 2018*). Adding further complexity, *CKIδ1 and CKIδ2* isoform expression patterns appear to be tissue specific and are regulated by m6A RNA modification. Regulation of CKI levels and isoform expression is an important direction for future work in the circadian clock.

CKI has a diverse array of functions in eukaryotes and is critically important in human health (reviewed in: *Cheong and Virshup, 2011*; *Vielhaber and Virshup, 2001*). CKI overexpression is pathogenic in Alzheimer's disease in addition to its role in circadian period regulation (*Sundaram et al., 2019*). Mutation of hPER2 at residue S662 is associated with the human sleep and circadian disorder FASPS (*Toh et al., 2001*), and CKIδ/CKIε kinases control the phosphorylation state of this critical site as well as phospho-switch regions dictating PER2 stability (*Narasimamurthy et al., 2018*; *Philpott et al., 2020*; *Zhou et al., 2015*). Significantly, mutation of human CKIδ itself phenocopies this, also leading to FASPS (*Xu et al., 2005*). Future work on the regulation of CKI levels and isoform expression will shed light on CKI regulation in the clock, in development, and in human disease.

# Materials and methods

## Key resources table

| Reagent type (species) or resource | Designation | Source or reference | Identifiers | Additional information |
|---|---|---|---|---|
| Gene (*Neurospora crassa*) | *prd-2* | FungiDB | NCU01019 | |
| Gene (*Neurospora crassa*) | *upf1prd-6* | FungiDB | NCU04242 | |
| Gene (*Neurospora crassa*) | *ck-1a* | FungiDB | NCU00685 | |
| Strain, strain background (*Neurospora crassa*) | *Supplementary file 1* | This study; Fungal Genetics Stock Center (FGSC) | | |
| Antibody | Anti-V5 (mouse monoclonal) | ThermoFisher | Cat. # R960-25 | (1:3000) |
| Antibody | Anti-tubulin alpha (mouse monoclonal) | Fitzgerald | Cat. # 10R-T130a | (1:10,000) |
| Antibody | Anti-CKI (rabbit polyclonal) | Generous gift from Michael Brunner (University of Heidelberg) | | (1:1000) |
| Antibody | Anti-FLAG M2 magnetic beads (mouse monoclonal) | Sigma | Cat. # M8823 | 30 µl beads incubated with 10 mg total protein for UV-CLIP |
| Recombinant DNA reagent | c box-luc (plasmid-derived construct) | As described, PMID:25635104 | | <500 bp of the *frq* promoter driving codon-optimized luciferase; targeted to the *csr-1* locus for selection |
| Chemical compound, drug | D-quinic acid | Sigma | Cat. # 138622 | 1 M stock solution, pH adjusted to 5.8 with NaOH |

*Continued on next page*

*Continued*

| Reagent type (species) or resource | Designation | Source or reference | Identifiers | Additional information |
|---|---|---|---|---|
| Chemical compound, drug | Allele-In-One Mouse Tail Direct Lysis Buffer | Allele Biotechnology | Cat. # ABP-PP-MT01500 | 50 μl reagent mixed with Neurospora asexual spores for gDNA isolation |
| Chemical compound, drug | Thiolutin | Cayman Chemical | Cat. # 11350 | Stock solution prepared in DMSO |
| Software, algorithm | Custom R software | https://github.com/cmk35 | | UTR length analyses from *Figure 5—figure supplement 2* |

## Neurospora strains and growth conditions

The *ras-1*$^{bd}$ *prd-2*$^{INV}$ strains 613–102 (*mat* A) and 613–43 (*mat* a) were originally isolated in the Feldman laboratory (*Lewis, 1995*). Strains used in this study were derived from the wild-type background (FGSC2489 *mat* A), *ras-1*$^{bd}$ background (87–3 *mat* a or 328–4 *mat* A), Δ*mus-51* background (FGSC9718 *mat* a), or the Fungal Genetics Stock Center (FGSC) knockout collection as indicated (*Supplementary file 1*). Strains were constructed by transformation or by sexual crosses using standard Neurospora methods (http://www.fgsc.net/Neurospora/NeurosporaProtocolGuide.htm).

The 'c box-luc' core clock transcriptional reporter was used to assay circadian period length by luciferase (*Figure 2B*, *Figure 4B*, *Figure 5A*, *Figure 1—figure supplement 1*, and *Figure 5—figure supplement 1*). In this construct, a codon-optimized firefly luciferase gene is driven by the clock box in the *frequency* promoter (*Gooch et al., 2008*; *Hurley et al., 2014*; *Larrondo et al., 2015*). The clock reporter construct was targeted to the *csr-1* locus and selected on resistance to 5 μg/ml cyclosporine A (Sigma # 30024) (*Bardiya and Shiu, 2007*).

Standard race tube (RT) medium was used for all RTs (1× Vogel's Salts, 0.1% glucose, 0.17% arginine, 1.5% agar, and 50 ng/ml biotin). Where indicated, D-quinic acid (Sigma # 138622) was added from a fresh 1 M stock solution (pH 5.8). Standard 96-well plate medium was used for all camera runs (1× Vogel's Salts, 0.03% glucose, 0.05% arginine, 1.5% agar, 50 ng/ml biotin, and 25 μM luciferin from GoldBio # 115144-35-9). Liquid cultures were started from fungal plugs as described (*Chen et al., 2009*; *Nakashima, 1981*) or from a conidial suspension at 1 × 10$^5$ conidia/ml. Liquid cultures were grown in 2% glucose liquid culture medium (LCM; 1× Vogel's Salts, 0.5% arginine w/v) or in 1.8% glucose Bird Medium (*Metzenberg, 2004*) as indicated. QA induction experiments in liquid culture were performed in 0.1% glucose LCM medium with QA supplemented. All the experiments were conducted at 25°C in constant light unless otherwise indicated.

Strains were genotyped by screening for growth on selection medium (5 μg/ml cyclosporine A, 400 μg/ml Ignite, and/or 200–300 μg/ml Hygromycin). PCR genotyping was performed on gDNA extracts from conidia incubated with Allele-In-One Mouse Tail Direct Lysis Buffer (Allele Biotechnology # ABP-PP-MT01500) according to the manufacturer's instructions. GreenTaq PCR Master Mix (ThermoFisher # K1082) was used for genotyping. Relevant genotyping primers for key strains are: *ras-1*$^{bd}$ (mutant): 5' TGCGCGAGCAGTACATGCGAAT and 5' CCTGATTTCGCGGACGAGATCGTA 3'; *ras-1*$^{WT}$ (NCU08823): 5' GCGCGAGCAGTACATGCGGAC 3' and 5' CCTGATTTCGCGGACGAGATCGTA 3'; *prd-2*$^{WT}$ (NCU01019): 5' CACTTCCAGTTATCTCGTCAC 3' and 5' CACAACCTTGTTAGGCATCG 3'; Δ*prd-2*::bar$^R$ (KO mutant): 5' CACTTCCAGTTATCTCGTCAC 3' and 5' GTGCTTGTCTCGATGTAGTG 3'; *prd-2*$^{INV}$ (left breakpoint): 5' AGCGAGCTGATATGCCTTGT 3' and 5' CGACTTCCACCACTTCCAGT 3'; *prd-2*$^{INV}$ (right breakpoint): 5' TGTTTGTCCGGTGAAGATCA 3' and 5' GTCGTGGAATGGGAAGACAT 3'; Δ*upf1*$^{prd-6}$::hyg$^R$ (FGSC KO mutant): 5' CTGCAACCTCGGCCTCCT 3' and 5' CAGGCTCTCGATGAGCTGATG 3'; bar$^R$::P$_{qa-2}$-*ck-1a* (QA inducible CKI): 5' GTGCTTGTCTCGATGTAGTG 3' and 5' GATGTCGCGGTGGATGAACG 3'.

## RNA stability assays

Control and Δ*prd-2* liquid cultures grown in 1.8% glucose Bird medium were age-matched and circadian time (CT) matched to ensure that RNA stability was examined at the same phase of the clock. Control cultures were shifted to constant dark for 12 hr, and Δ*prd-2* cultures were shifted to dark for 14 hr (~CT1 for 22.5 hr wild-type period and for 26 hr Δ*prd-2* period; 46 hr total growth). Thiolutin (THL; Cayman Chemical # 11350) was then added to a final concentration of 12 µg/ml to inhibit new RNA synthesis. Samples were collected every 10 min after THL treatment by vacuum filtration and flash frozen in liquid nitrogen. THL has multiple off-target effects in addition to inhibiting transcription (*Lauinger et al., 2017*). For this reason, *frq* mRNA degradation kinetics were also examined with an alternative protocol. Light-grown, age-matched liquid Bird cultures of wild-type and Δ*prd-2* were shifted into the dark and sampled every 10 min to measure *frq* turnover; transcription of *frq* ceases immediately on transfer to darkness (*Heintzen et al., 2001*; *Tan et al., 2004*). All tissue manipulation in the dark was performed under dim red lights, which do not reset the Neurospora clock (*Chen et al., 2009*).

## RNA isolation and detection

Frozen Neurospora tissue was ground in liquid nitrogen with a mortar and pestle. Total RNA was extracted with TRIzol (Invitrogen # 15596026) and processed as described (*Chen et al., 2009*). RNA samples were prepared for RT-qPCR, northern blotting, RNA-sequencing, or stored at −80˚C.

For RT-qPCR, cDNA was synthesized using the SuperScript III First-Strand synthesis kit (Invitrogen # 18080–051). RT-qPCR was performed using SYBR green master mix (Qiagen # 204054) and a StepOne Plus Real-Time PCR System (Applied Biosystems). $C_t$ values were determined using StepOne software (Life Technologies) and normalized to the *actin* gene ($\Delta C_t$). The $\Delta\Delta C_t$ method was used to determine mRNA levels relative to a reference time point. Relevant RT-qPCR primer sequences are: *prd-2* (NCU01019): 5' GGGCAACGACGTCAAACTAT 3' and 5' TGCGTGTACATCACTCTGGA 3', and *actin* (NCU04173): 5' GGCCGTGATCTTACCGACTA 3' and 5' TCTCCTTGATGTCACGAACG 3'.

Northern probes were first synthesized using the PCR DIG Probe Synthesis Kit (Roche # 11 636 090 910). The 512 bp *frq* probe was amplified from wild-type Neurospora genomic DNA with primers: 5' CTCTGCCTCCTCGCAGTCA 3' and 5' CGAGGATGAGACGTCCTCCATCGAAC 3'. The 518 bp *ck-1a* probe was amplified with primers: 5' CCATGCCAAGTCGTTCATCC 3' and 5' CGGTCCAG TCAAAGACGTAGTC 3'. Total RNA samples were prepared according to the NorthernMax-Gly Kit instructions (Invitrogen # AM1946). Equal amounts of total RNA (5–10 µg) were loaded per lane of a 0.8–1% w/v agarose gel. rRNA bands were visualized prior to transfer to validate RNA integrity. Transfer was completed as described in the NorthernMax-Gly instructions onto a nucleic acid Amersham Hybond-N+ membrane (GE # RPN303B). Transferred RNA was cross-linked to the membrane using a Stratalinker UV Crosslinker. The membrane was blocked and then incubated overnight at 42˚ C in hybridization buffer plus the corresponding DIG probe. After washing with NorthernMax-Gly Kit reagents, subsequent washes were performed using the DIG Wash and Block Buffer Set (Roche # 11 585 762 001). Anti-Digoxigenin-AP Fab fragments were used at 1:10,000. Chemiluminescent detection of anti-DIG was performed using CDP-Star reagents from the DIG Northern Starter Kit (Roche # 12 039 672 910). Densitometry was performed in ImageJ.

Total RNA was submitted to Novogene for stranded polyA+ library preparation and sequencing. 150 bp paired-end (PE) read libraries were prepared, multiplexed, and sequenced in accordance with standard Illumina HiSeq protocols. 24.8 ± 1.7 million reads were obtained for each sample. Raw FASTQ files were aligned to the *Neurospora crassa* OR74A NC12 genome (accessed September 28, 2017, via the Broad Institute: ftp://ftp.broadinstitute.org/pub/annotation/fungi/neurospora_crassa/assembly/) using STAR (*Dobin et al., 2013*). On average, 97.6 ± 0.3% of the reads mapped uniquely to the NC12 genome. Aligned reads were assembled into transcripts, quantified, and normalized using Cufflinks2 (*Trapnell et al., 2013*). Triplicate control and Δ*prd-2* samples were normalized together with CuffNorm, and the resulting FPKM output was used in the analyses presented. RNA-sequencing data have been submitted to the NCBI Gene Expression Omnibus (GEO; https://www.ncbi.nlm.nih.gov/geo/) under accession number GSE155999.

## CLIP assay

CLIP was performed using PUF4 (NCU16560) as a positive control RNA-binding protein from *Wilinski et al., 2017*, with modifications. Neurospora strains containing endogenous locus C-terminally VHF tagged PUF4, PRD-2, or untagged negative control were used (*Supplementary file 1*). Liquid cultures were grown in 2% glucose LCM for 48 hr in constant light. Tissue was harvested by vacuum filtration and fixed by UV cross-linking for 7 min on each side of the fungal mat (Stratalinker UV Crosslinker 1800 with 254 nm wavelength bulbs). UV cross-linked tissue was frozen in liquid nitrogen and ground into a fine powder with a mortar and pestle. Total protein was extracted in buffer (25 mM Tris-HCl pH 7.4, 150 mM NaCl, 2 mM $MgCl_2$, 0.5% NP-40, 1 mM DTT, $1\times$ cOmplete protease inhibitor, 100 U/ml RNAse Out) and concentration determined by Bradford Assay (Bio-Rad # 500–0006). Approximately 10 mg of total protein was added to 30 µl anti-FLAG M2 magnetic beads (Sigma # M8823) prepared according to the manufacturer's instructions. Beads and lysate were rotated for 4 hr at 4°C, followed by four washes in 750 µl extraction buffer (5–10 min rotating per wash). Bound RNA-binding proteins were eluted with 100 µl 0.1 M glycine-HCl pH 3.0 for 10 min. The supernatant was collected using a magnetic rack (NEB S1506S) and neutralized in 10 µl of 1 M Tris-HCl pH 8.0. The elution was incubated with 300 µl of TRIzol (Invitrogen # 15596026) for 10 min to extract RNA. Total RNA was isolated, DNAse treated, and concentrated using the Direct-zol RNA Microprep Kit (Zymo # R2062) following the manufacturer's instructions.

Equal amounts of immunoprecipitated RNA (~50 ng) were converted into cDNA using the oligo (dT) method from the SuperScript IV First-Strand synthesis kit (Invitrogen # 18091–050). RT-qPCR was performed using SYBR green master mix (Qiagen # 204054) and a StepOne Plus Real-Time PCR System (Applied Biosystems). $C_t$ values were determined using StepOne software (Life Technologies) and normalized to the *crp-43* gene ($\Delta C_t$) instead of the *actin* (NCU04173) gene because *actin* is a putative PUF4 target by HITS-CLIP (*Wilinski et al., 2017*). The $\Delta\Delta C_t$ method was used to determine target mRNA enrichment relative to the negative IP sample. Relevant RT-qPCR primer sequences were designed to flank introns: *cbp3* (NCU00057; PUF4 target): 5' CGAGAAATTCGGCCTTCTCCC 3' and 5' GCCTGGTGGAAGAAGTGGT 3'; *mrp-1* (NCU07386; PUF4 target): 5' TAGTAGGCACCGACTTTGAGCA 3' and 5' CGGGGACAGGTGGTCGAA 3'; *ck-1a* (NCU00685; PRD-2 target): 5' CGCAAACATGACTACCATG 3' and 5' CTCTCCAGCTTGATGGCA 3'; *crp-43* (NCU08964; normalization control): 5' CTGTCCGTACTCGTGACTCC 3' and 5' ACCATCGATGAGGAGCTTGC 3'.

## Protein isolation and detection

Frozen Neurospora tissue was ground in liquid nitrogen with a mortar and pestle. Total protein was extracted in buffer (50 mM HEPES pH 7.4, 137 mM NaCl, 10% glycerol v/v, 0.4% NP-40 v/v, and cOmplete Protease Inhibitor Tablet according to instructions for Roche # 11 836 170 001) and processed as described (*Garceau et al., 1997*). Protein concentrations were determined by Bradford Assay (Bio-Rad # 500–0006). For western blots, equal amounts of total protein (10–30 µg) were loaded per lane into 4–12% Bis-Tris Bolt gels (Invitrogen # NW04125BOX). Western transfer was performed using an Invitrogen iBlot system (# IB21001) and PVDF transfer stack (# IB401001). Primary antibodies used for western blotting were anti-V5 (1:3000, ThermoFisher # R960-25), anti-Tubulin alpha (1:10,000, Fitzgerald # 10R-T130a), or anti-CK1a (1:1000, rabbit raised). The secondary antibodies, goat anti-mouse or goat anti-rabbit HRP, were used at 1:5000 (Bio-Rad # 170–6516, # 170–6515). SuperSignal West Pico PLUS Chemiluminescent Substrate (ThermoFisher # 34578) or Femto Maximum Sensitivity Substrate (ThermoFisher # 34095) was used for detection. Immunoblot quantification and normalization were performed in ImageJ.

Nuclear and cytosolic fractions were prepared as previously described (*Hong et al., 2008*). Approximately 10 µg of total protein from each fraction was loaded for immunoblotting. Primary antibodies for fraction controls were histone H3A (Fitzgerald) and γ-tubulin (Abcam). HRP-conjugated secondary antibodies (Bio-Rad) were used with SuperSignal West Pico ECL (Thermo) for detection.

## Luciferase reporter detection and data analysis

96-well plates were inoculated with conidial suspensions from strains of interest and entrained in 12 hr light:dark cycles for 2 days in a Percival incubator at 25°C. Temperature inside the Percival incubator was monitored using a HOBO logger device (Onset # MX2202) during entrainment and free run.

Plates were then transferred into constant darkness to initiate the circadian free run. Luminescence was recorded using a Pixis 1024B CCD camera (Princeton Instruments). Light signal was acquired for 10–15 min every hour using LightField software (Princeton Instruments, 64-bit version 6.10.1). The average intensity of each well was determined using a custom ImageJ Macro (*Larrondo et al., 2015*), and background correction was performed for each frame. Results from two different algorithms were averaged together to determine circadian period from background-corrected luminescence traces. The MESA algorithm was used as previously described (*Kelliher et al., 2020*). A second period measurement was obtained from an ordinary least squares autoregressive model to compute the spectral density (in R: spec.ar(. . ., method='ols')). RT period lengths were measured from scans using ChronOSX 2.1 software (*Roenneberg and Taylor, 2000*).

### Data visualization

All figures were plotted in R, output as scalable vector graphics, formatted using Inkscape, and archived in R markdown format. Data represent the mean of at least three biological replicates with standard deviation error bars, unless otherwise indicated.

## Acknowledgements

We thank the Fungal Genetics Stock Center (Kansas City, Missouri, USA) for curating *N. crassa* strains. We thank Arun Mehra for discussions on preliminary work to identify the clock-relevant mechanism of the *upf1^prd-6* mutation, Bin Wang for assistance in constructing and validating the CKI^SHORT hyperactive allele, Jill Emerson for assistance in constructing Δ*prd-2* (NCU01019), and Brad Bartholomai for discussions on *prd-2*. We acknowledge Jerry Feldman for advice on the *upf1^prd-6* gene naming convention. The Neurospora CK1a antibody was courtesy of Michael Brunner (University of Heidelberg). This work was supported by the National Institutes of Health (F32 GM128252 to CMK, R35 GM118021 to JCD, R35 GM118022 to JJL) and EMSL (50173 to Co-PI JCD).

## Additional information

### Funding

| Funder | Grant reference number | Author |
| --- | --- | --- |
| National Institutes of Health | F32 GM128252 | Christina M Kelliher |
| National Institutes of Health | R35 GM118021 | Jay C Dunlap |
| National Institutes of Health | R35 GM118022 | Jennifer J Loros |
| EMSL | 50173 | Jay C Dunlap |

The funders had no role in study design, data collection and interpretation, or the decision to submit the work for publication.

### Author contributions

Christina M Kelliher, Conceptualization, Resources, Data curation, Formal analysis, Funding acquisition, Validation, Investigation, Visualization, Methodology, Writing - original draft, Writing - review and editing; Randy Lambreghts, Qijun Xiang, Christopher L Baker, Resources, Investigation, Methodology; Jennifer J Loros, Jay C Dunlap, Conceptualization, Supervision, Funding acquisition, Writing - original draft, Writing - review and editing

### Author ORCIDs

Christina M Kelliher http://orcid.org/0000-0002-4554-1818
Jay C Dunlap https://orcid.org/0000-0003-1577-0457

### Decision letter and Author response

Decision letter https://doi.org/10.7554/eLife.64007.sa1
Author response https://doi.org/10.7554/eLife.64007.sa2

## Additional files

### Supplementary files

• Supplementary file 1. *Neurospora crassa* strains used in this study.

• Transparent reporting form

### Data availability

RNA-Sequencing data have been deposited in GEO under accession GSE155999.

The following dataset was generated:

| Author(s) | Year | Dataset title | Dataset URL | Database and Identifier |
|---|---|---|---|---|
| Kelliher CM, Lambreghts R, Xiang Q, Baker CL, Loros JJ, Dunlap JC | 2020 | Nonsense mediated decay and a novel protein Period-2 regulate casein kinase I in an opposing manner to control circadian period in Neurospora crassa | https://www.ncbi.nlm. nih.gov/geo/query/acc. cgi?acc=GSE155999 | NCBI Gene Expression Omnibus, GSE155999 |

The following previously published dataset was used:

| Author(s) | Year | Dataset title | Dataset URL | Database and Identifier |
|---|---|---|---|---|
| Zhang Y, Guo J | 2017 | RNA-seq analysis of wild type and upf1 knockout strains in the filamentous fungus Neurospora crassa | https://www.ncbi.nlm. nih.gov/geo/query/acc. cgi?acc=GSE97157 | NCBI Gene Expression Omnibus, GSE97157 |

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
