## [Decision Letter]

**Acceptance summary:**

The work teaches us how a mutation affecting nonsense mediated mRNA decay (NMD) causes a change in period length of the circadian clock in *Neurospora crassa*. The work elegantly demonstrates how a specific RNA binding protein binds to the transcript of the central clock regulator CKI and protects it from NMD.

**Decision letter after peer review:**

Thank you for submitting your article "PERIOD-2 directly regulates *casein kinase I* and counteracts nonsense mediated decay in the *Neurospora* circadian clock" for consideration by *eLife*. Your article has been reviewed Detlef Weigel as Reviewing and Senior Editor and three reviewers. The following individuals involved in review of your submission have agreed to reveal their identity: Matthew S. Sachs (Reviewer #2).

I am happy to say that the reviews were uniformly positive. To paraphrase some of the comments: You are providing a convincing answer to a longstanding question concerning how a mutation affecting nonsense mediated mRNA decay (NMD) in *Neurospora crassa* causes a change in period length of the circadian clock. You explain this effect by revealing that the CKI gene encoding a conserved, central clock regulator, casein kinase I, is regulated at the level of RNA stability by the NMD mechanism. Specifically, you elegantly demonstrate that the prd-2 mutation affects an RNA binding protein that binds to the CKI transcript and protects it from NMD. These are exciting findings that are conveyed in a clear and cohesive manner.

I am including the three reviews in full below. As you will see, the reviewers had several major points:

1) Because of the PERIOD (PER) gene in animals has the same name as period (per) loci in Neurocrassa, it is important to make clear that per-2 is a completely different locus than one of the PER loci (especially since there is a PER2 in vertebrates. As a dyed-in-the-wool geneticist I believe in the primacy of genetics and am a fan of original gene names, but in this specific case, I would encourage you to rename per-2 as upf1.

2) In addition, an even tighter focus on the clock in general would make the work more accessible to *eLife*'s broad audience.

3) The reviewers also suggest a couple of experiments for the future; should you already have relevant data you wish to include I would invite you to do so. If you mention this, please do so prominently in the cover letter of the revision, so that I can have a look at them myself, since I am not intending to send this out for re-review.

4) Finally, I am happy that we had prior to this submission a Discussion of what would be needed to make the work a likely *eLife* candidate, and I am delighted that this has worked out well.

Reviewer #1:

This manuscript presents the identification of the *Neurospora* period-2 allele and reveals its role in stabilization of the mRNA for the core clock gene casein kinase 1a (ck1a). This discovery fits in nicely with prior observations of another period allele, prd-6, that promotes degradation of ck1a mRNA by nonsense-mediated decay (NMD) to collectively implicate control of ck1a mRNA stability as a critical point in circadian period determination in Neurospora. Overall, the study provides a balanced and rigorous experimental approach and thoughtfully describes the findings in the context of the *Neurospora* clock and functionally related clocks in other organisms. This is an exciting finding conveyed in a clear and cohesive manner that warrants publication in *eLife*.

Reviewer #2:

Kelliher et al., convincingly answer a longstanding question concerning how a mutation affecting nonsense mediated mRNA decay (NMD) in *Neurospora crassa* causes a change in the period length of the organism's circadian rhythm. They identify a crucial clock-control gene, casein kinase I, as the RNA-level target of NMD. Furthermore, they determine that the previously uncharacterized circadian mutant prd-2, encodes an RNA binding protein that binds to the CKI transcript and protects it from NMD. Their data provide a molecular explanation for formal genetic epistasis relationships that have gone unexplained for decades and furthermore provide additional bases for considering the role of NMD in maintaining circadian rhythms in other eukaryotes.

Essential revisions:

1) Readers will likely wonder why the regions defining the PRD2 binding site(s) on the CK1 transcript were not more precisely identified. There is discussion concerning lack of conserved sequences (Figure 3—figure supplement 1 legend) but this point is not fully clarified for CKI which is demonstrated by CLIP to interact with PRD2.

2) Looking at Querforth, 2007 (their Figure 1A, iii and iv): the short CKI form would be produced from transcripts containing 3'UTR introns and would be predicted to be subject to EJC-mediated NMD and to long 3'UTR NMD; their Figure 1A i and ii transcripts would be subject to long 3'UTR NMD but not to EJC-mediated NMD. I think these points could be elaborated in terms of how CKI transcripts could be degraded by NMD – they may be relevant to PRD2 mechanism of protection.

Reviewer #3:

This manuscript identifies the genetic basis of circadian phenotypes in two long-described mutants of *Neurospora crassa*, Period-2 and Period-6, as defects in regulating the stability of the ck1a transcript (encoding Casein Kinase I). The period-2 mutant identifies a locus that encodes an RNA binding protein that stabilizes the ck1a message; in the mutant the stabilizing protein is absent, ck1a levels are low, and circadian period is long. Experiments demonstrated that the low level of ck1a is sufficient to explain the phenotype. Genetic experiments indicated that the period-2 mutant is in the same pathway as period-6, which was known to encode a component of nonsense-mediated decay. They tested some hypotheses, which proved correct, that the relevant target in period-6 is also ck1a. In the period-6 mutant CK1 levels are too high. The experiments are thoughtful and thorough, and clearly presented. Concerns about the manuscript relate to presentation and suitability for a broad audience.

1) Although the mutant is named period-2, the gene and protein should be given a functionally related name that won't be confused with period (per), the single most famous gene in the circadian rhythms field. Particularly period2 (per2), is the most clock-impactful of the three mammalian period genes. It is difficult enough for those outside the circadian field to navigate the components without having the same name, albeit with a different abbreviation, used for non-homologous components.

2) The approachability for a broad audience is also diminished by inclusion in the introduction of details related to the *Neurospora* clock and specifically text to draw parallels to the clocks of animals (similar in outline but not homologous) that are not germane to this paper. Specifically, lines 60-74 are off-topic details (antisense RNAs to other components, intrinsic disorder) that will bog down someone outside the field. Keeping a clean focus on the role of CK1 will improve the accessibility. The universality of CK1 in the clock across kingdoms is appropriately highlighted, although some of the information along these lines in the introduction would be better moved to the Discussion.

3) One aspect of parallels drawn to the animal clock is misleading and should be reworded or removed. In line 55, FRQ is said to be functionally homologous to PERs and CRYs. The statement is not meaningful, because homology is an evolutionary term and the three proteins in question are not in any way phylogenetically conserved, and they are not even analogs, as their molecular mechanisms are not the same. They are similar in acting to oppose the activity of transcription factors. Implying something more closely linked at the molecular level is not correct.

---

## [Author Response]

1) Because of the PERIOD (PER) gene in animals has the same name as period (per) loci in Neurocrassa, it is important to make clear that per-2 is a completely different locus than one of the PER loci (especially since there is a PER2 in vertebrates. As a dyed-in-the-wool geneticist I believe in the primacy of genetics and am a fan of original gene names, but in this specific case, I would encourage you to rename per-2 as upf1.

This is an excellent suggestion for reader clarity. We have relabeled *prd-6* as *upf1^prd-6^* throughout (see subsection “An Interstitial Inversion Identifies *prd”*). As geneticists, we share the visceral reaction to suggestions about gene renaming, and so we have also corresponded with Dr. Jerry Feldman in whose laboratory the *prd* genes were originally isolated and named. Dr. Feldman agreed that *upf1^prd-6^*is a suitable naming convention. Furthermore, one sentence has been added in subsection “An Interstitial Inversion Identifies *prd”* to clarify that the commonly known mammalian and insect PERIOD gene(s) is not among the classical *Neurospora period* (*prd*) mutants. In response to reviewer 1 comment # 4, we have also added text in subsection “Nonsense Mediated Decay Impacts the Clock by Regulating CKI Levels” detailing the lack of obvious evolutionary conservation of PRD-2 beyond ascomycete fungi. Taken together, we believe that these changes will further clarify that *Neurospora prd-6* encodes UPF1, a conserved NMD component, that *Neurospora prd-2* encodes a fungal-specific RNA-binding protein, and that both PRDs are distinct from the animal PER proteins. In addition, the title now refers to this novel fungal gene by its three letter mnemonic so there should be no basis for confusion.

2) In addition, an even tighter focus on the clock in general would make the work more accessible to eLife's broad audience.

We agree completely on the importance of reader accessibility. In particular, we followed the advice of reviewer # 3 to re-structure the text in the Introduction and Discussion section (comments # 2 – 3). The first two paragraphs of the Introduction were simplified to highlight the most important conserved features of the negative arm of the circadian clock, which are relevant to this work and important introductory material. Text and citations describing more specific examples of conserved post-transcriptional regulation of the negative arm were moved to a new paragraph in the Discussion. A pair of sentences describing Casein Kinase I’s central role in a sleep disorder called FASPS were moved from the Introduction to the Discussion paragraph on CKI in human health. We hope that these (largely organizational) changes have improved the accessibility of this work to our audience.

3) The reviewers also suggest a couple of experiments for the future; should you already have relevant data you wish to include, I would invite you to do so. If you mention this, please do so prominently in the cover letter of the revision, so that I can have a look at them myself, since I am not intending to send this out for re-review.

As mentioned in the text above, please find new data in Figure 3–Figure supplement 2B and Figure 5figure –figure supplement 1 in response to reviewer comments.

Figure 3–Figure supplement 2 shows, by two different experimental methods, that *frequency* (*frq*) mRNA stability is not significantly altered in the D*prd-2* RNA-binding protein mutant, unlike the drastic decrease in stability observed for *casein kinase I* (Figure 3B). The new experiment demonstrates that *frq* RNA turnover is similar in control and D*prd-2* after chemical inhibition of RNA synthesis by thiolutin. This result also serves as a control for *casein kinase I* instability in D*prd-2* after thiolutin treatment shown in Figure 3B, as correctly pointed out by reviewer 2 in comment # 10 below.

As described in the main text (–Discussion section), nonsense-mediated decay (NMD) is triggered by two categories of RNA features. mRNAs that contain an intron(s) in the 3’ UTR are degraded by the NMD machinery together with the exon junction complex (EJC) and nuclear cap-binding complex (CBC). mRNAs that contain 5’ UTR uORFs and/or long 3’ UTRs are degraded by the NMD subunits alone (UPF1, UPF2, and UPF3). Reviewer #2 correctly points out that *ck-1a* transcripts have features of both NMD-targeting mechanisms and, reasonably, asks which pathway predominantly regulates *ck-1a* turnover (Comments # 2, 3, 7, 8). The new Figure 5–Figure supplement 1 provides evidence that only the NMD machinery is clock-relevant and used to regulate *ck-1a* levels. Using a CBC component knockout D*cbp80*, we show that (1) the nuclear cap-binding complex is not required for normal circadian period length, and (2) there is no genetic interaction between D*prd-2* (unstable *ck-1a*, long period) and D*cbp80*. Given the result in Figure 5A where all three individual NMD subunits are both short circadian period length (18 – 20 hours) and genetically epistatic to D*prd-2*, we propose that only the NMD machinery is required to regulate *ck-1a*.

4) Finally, I am happy that we had prior to this submission a Discussion of what would be needed to make the work a likely eLife candidate, and I am delighted that this has worked out well.

We are equally pleased with the outcome of our discussions on revisiting the first submission of this manuscript. The RNA-binding experiments suggested by you not only greatly improved the story and reception of this current manuscript, but also will be an invaluable technique for the fungal genetics community in the future. We are grateful for your time and consideration.

Reviewer #1:This manuscript presents the identification of the Neurospora period-2 allele and reveals its role in stabilization of the mRNA for the core clock gene casein kinase 1a (ck1a). This discovery fits in nicely with prior observations of another period allele, prd-6, that promotes degradation of ck1a mRNA by nonsense-mediated decay (NMD) to collectively implicate control of ck1a mRNA stability as a critical point in circadian period determination in Neurospora. Overall, the study provides a balanced and rigorous experimental approach and thoughtfully describes the findings in the context of the Neurospora clock and functionally related clocks in other organisms. This is an exciting finding conveyed in a clear and cohesive manner that warrants publication in eLife.While the Guo et al. (2019) paper cited on lines 101 and 460 has some data to suggest that autophosphorylaMon might parMcipate in Mming of the mammalian circadian Mming, it did not make the fundamental discovery of CK1 autophosphorylaMon; perhaps one or more of the original papers that discovered CK1 autophosphorylaMon and its role in controlling kinase acMvity in cells should be cited to provide recogniMon of that work (e.g., Vielhaber, E. et al. 1998 JBC, Rivers, A. et al. 1998 JBC, or Gietzen, K and Virshup 1999 JBC).

We apologize for this literature oversight and agree completely that including a citation for the original work on CKI autophosphorylation best serves the reader. After reviewing the suggested literature, Gietzen and Virshup 1999 JBC has been added as a reference for functional characterization of CKIe C-terminal tail truncations (and identification of 8 key Ser/Thr residues within the C-terminus) in the kinase’s inhibitory autophosphorylation. Guo *et al.* 2019 JBR is retained as a reference to convey the continued relevance and lack of complete understanding of CKI regulation in circadian biology.

Reviewer #2:Kelliher et al., convincingly answer a longstanding question concerning how a mutation affecting nonsense mediated mRNA decay (NMD) in *Neurospora crassa* causes a change in the period length of the organism's circadian rhythm. They identify a crucial clock-control gene, casein kinase I, as the RNA-level target of NMD. Furthermore, they determine that the previously uncharacterized circadian mutant prd-2, encodes an RNA binding protein that binds to the CKI transcript and protects it from NMD. Their data provide a molecular explanation for formal genetic epistasis relationships that have gone unexplained for decades and furthermore provide additional bases for considering the role of NMD in maintaining circadian rhythms in other eukaryotes.Substantive concerns:1) Readers will likely wonder why the regions defining the PRD2 binding site(s) on the CK1 transcript were not more precisely identified. There is discussion concerning lack of conserved sequences (Figure 3—figure supplement 1 legend) but this point is not fully clarified for CKI which is demonstrated by CLIP to interact with PRD2

This is an excellent observation and identification of the sequence motif(s) within the *ck-1a* 3’ UTR underlying its regulation by PRD-2 and by NMD will be the focus of future studies. Here is a full description of our informatic attempts to identify PRD-2 sequence motifs, summarized in Figure 3—figure supplement 1 legend. As described in Figure 3–figure supplement 1, we identified 292 genes downregulated significantly in D*prd-2* by RNA-Seq and hypothesized that, if PRD-2 acts to stabilize its targets, this group of downregulated genes contains *bona fide* PRD-2 targets. Of the 292 downregulated genes, 226 genes were annotated with 3’ UTR sequence coordinates and 212 genes with 5’ UTRs. Focusing specifically on 3’ UTRs due to importance in *ck-1a* regulation, we ran Weeder2 to identify sequence motifs within the top 30, top 50, top 70, top 100, or all 226 3’ UTRs from D*prd-2* downregulated genes. Zero motifs emerged with a Weeder2 score > 1.5. For reference, the known binding motif “GATCG” for WCC is recovered at a Weeder2 score = 2.01 from the top 33 WCC targets by ChIP-seq, and the known binding motif “GGACCCT” for CSP-1 is recovered at a Weeder2 score = 2.44 from over 600 known targets by ChIP-seq. Inspecting the motifs scoring better than 1.3, we did identify one motif “GGATATA” among the D*prd-2* downregulated gene UTRs with Weeder2 scores ranging from 1.33 – 1.37 (depending on the input sequences). Curiously, a similar motif “GGATAT” does appear twice toward the end of the 3’ UTR in *ck-1a*, and this could be a clue to PRD-2 binding despite its relatively poor sequence enrichment score among many sequences. In future work, we will delete regions of the *ck-1a* 3’ UTR (including the putative PRD-2 motifs) and screen for circadian period defects.

2) Looking at Querforth, 2007 (their Figure 1A, iii and iv): the short CKI form would be produced from transcripts containing 3'UTR introns and would be predicted to be subject to EJC-mediated NMD and to long 3'UTR NMD; their Figure 1A i and ii transcripts would be subject to long 3'UTR NMD but not to EJC-mediated NMD. I think these points could be elaborated in terms of how CKI transcripts could be degraded by NMD – they may be relevant to PRD2 mechanism of protection.

We appreciate this detailed observation placing our findings on *ck-1a* regulation in the context of previous work on *Neurospora ck-1a* splicing patterns. We agree completely that only CKI transcript isoforms iii and iv would be subject to 3’ UTR intron mediated degradation by NMD, EJC, and CBC. The other two isoforms (i and ii) contain long 3’ UTRs only for NMD targeting. Of note, there are also two introns located in the 5’ UTR of *ck-1a* (source: FungiDB), which could produce additional isoforms and/or 5’ UTR uORFs for NMD-mediated targeted degradation.

Turning to our Figure 5F, we observe by visual inspection that both the long and the short isoforms of CKI protein increase in abundance in the NMD mutant (EJC-dependent and EJC-independent degradation both absent). However, if the CKI short isoform encoded by mRNAs iii and iv was the major CKI form subject to NMD regulation, we would not expect to see both protein isoforms increase in abundance in D*upf1^prd-6^*. Please see main comment # 3 for an additional experiment demonstrating that NMD machinery alone is required to regulate *ck-1a* in the context of the clock.

Reviewer #3:This manuscript identifies the genetic basis of circadian phenotypes in two long-described mutants of *Neurospora crassa*, Period-2 and Period-6, as defects in regulating the stability of the ck1a transcript (encoding Casein Kinase I). The period-2 mutant identifies a locus that encodes an RNA binding protein that stabilizes the ck1a message; in the mutant the stabilizing protein is absent, ck1a levels are low, and circadian period is long. Experiments demonstrated that the low level of ck1a is sufficient to explain the phenotype. Genetic experiments indicated that the period-2 mutant is in the same pathway as period-6, which was known to encode a component of nonsense-mediated decay. They tested some hypotheses, which proved correct, that the relevant target in period-6 is also ck1a. In the period-6 mutant CK1 levels are too high. The experiments are thoughtful and thorough, and clearly presented. Concerns about the manuscript relate to presentation and suitability for a broad audience.1) Although the mutant is named period-2, the gene and protein should be given a functionally related name that won't be confused with period (per), the single most famous gene in the circadian rhythms field. Particularly period2 (per2), is the most clock-impactful of the three mammalian period genes. It is difficult enough for those outside the circadian field to navigate the components without having the same name, albeit with a different abbreviation, used for non-homologous components.

This is an excellent point of clarification, which is also highlighted in main comment #1 above. One sentence has been added to the Results section stating explicitly that the *Neurospora prd* mutants are not related to the *period* gene(s) in fly or mammal. We hope that the use of the locus identifier “PRD” throughout (rather than “PER”) also alleviates some confusion between these distinct clock factors.

2) The approachability for a broad audience is also diminished by inclusion in the introduction of details related to the Neurospora clock and specifically text to draw parallels to the clocks of animals (similar in outline but not homologous) that are not germane to this paper. Specifically, the second paragraph of the Introduction is off-topic details (antisense RNAs to other components, intrinsic disorder) that will bog down someone outside the field. Keeping a clean focus on the role of CK1 will improve the accessibility. The universality of CK1 in the clock across kingdoms is appropriately highlighted, although some of the information along these lines in the Introduction would be better moved to the Discussion.

We appreciate this advice, which is also highlighted in main comment #2. We completely agree that the paragraph in question (originally located in the Introduction) contained too much detail on the regulation of FRQ, PER, and CRY, which is not the main thrust of this work on CKI regulation. This paragraph has been greatly shortened and simplified for reader approachability (Introduction). In the subsequent Introductory paragraph, we appreciate that the conserved nature of CKI in the clock was properly conveyed. To simplify this paragraph as suggested, we have moved two sentences on the human sleep and circadian disorder FASPS into the Discussion paragraph on human health.

3) One aspect of parallels drawn to the animal clock is misleading and should be reworded or removed. In the Introduction, FRQ is said to be functionally homologous to PERs and CRYs. The statement is not meaningful, because homology is an evolutionary term and the three proteins in question are not in any way phylogenetically conserved, and they are not even analogs, as their molecular mechanisms are not the same. They are similar in acting to oppose the activity of transcription factors. Implying something more closely linked at the molecular level is not correct.

We are in complete agreement that FRQ and PER/CRY are not evolutionarily related by sequence homology and that this point was not made clear enough in the original version of this manuscript. To address this, we have removed the statement in question “FRQ is functionally homologous to PERs and CRYs in the animal clock” and added an explicit statement that the negative arm proteins are not phylogenetically conserved between animals and fungi in the Introduction. However, we respectfully disagree that FRQ and PER/CRY only share the molecular and functional similarity of inhibiting the positive arm transcription factors (TFs). Both FRQ and PER/CRY nucleate the negative arm complex and recruit kinases such as Casein Kinases I and II to phosphorylate and inhibit the positive arm TFs. FRQ, PERs, and CRYs are heavily post-translationally modified, and in particular progressive phosphorylation seems to be a conserved timekeeping feature of the negative arm across species. Furthermore, FRQ, PER, and CRY all contain features of intrinsically disordered proteins (IDPs), which are thought to be under less selective evolutionary pressure due to lack of structural constraints (e.g. Brown et al., 2011). Thus, lack of sequence conservation cannot be taken as evidence of lack of evolutionary relatedness of function (e.g. Brody, 2020). Based on our current understanding of circadian evolution, though, we completely agree that language should be softened on relationships between FRQ, PERs, and CRYs. We hope that revisions to the Introduction and Discussion section (please see main comment #2) alleviate this concern with the original text.